



**Contribution of the nongrowing season to annual N₂O emissions from the**
**continuous permafrost region in Northeast China**
Weifeng Gao[1,2], Dawen Gao[1,2,3*], Liquan Song[1], Houcai Sheng[4,5], Tijiu Cai[4,5], Hong
Liang[1*]
[1]School of Environment and Energy Engineering, Beijing University of Civil
Engineering and Architecture, Beijing 100044, China
[2]Center for Ecological Research, Northeast Forestry University, Harbin 150040,
China
[3]State Key Laboratory of Urban Water Resource and Environment, Harbin Institute of
Technology, Harbin 150090, China
[4]School of Forestry, Northeast Forestry University, Harbin 150040, China
[5]Key Laboratory of Sustainable Forest Ecosystem Management-Ministry of Education,
Northeast Forestry University, Harbin 150040, China
**\*   Correspondence:**   Dawen   Gao   (gaodw@hit.edu.cn);   Hong   Liang
(liangh119@hit.edu.cn)
*E-mail address*: gaowf797@nenu.edu.cn (W. Gao); gaodw@hit.edu.cn (D. Gao);
songliquan@nefu.edu.cn   (L.   Song);   shenghoucai@163.com   (H.   Sheng);
caitijiu1963@163.com (T. Cai); liangh119@hit.edu.cn (H. Liang)



**Abstract.** Permafrost regions store large amounts of soil organic carbon and nitrogen, which are major sources of greenhouse gas. With climate warming, permafrost regions are thawing, releasing an abundance of greenhouse gases to the atmosphere and contributing to climate warming. Numerous studies have shown the mechanism of nitrous oxide ($N_2O$) emissions from the permafrost region during the growing season. However, little is known about the temporal pattern and drivers of nongrowing season $N_2O$ emissions from the permafrost region. In this study, $N_2O$ emissions from the permafrost region were investigated from June 2016 to June 2018 using the static opaque chamber method. Our aims were to quantify the seasonal dynamics of nongrowing season $N_2O$ emissions and its contribution to the annual budget. The results showed that the $N_2O$ emissions ranged from $-35.75$ to $74.16$ $\mu g \cdot m^{-2} \cdot h^{-1}$ during the nongrowing season in the permafrost region. The mean $N_2O$ emission from the growing season were $1.75$–$2.86$ times greater than that of winter and $1.31$–$1.53$ times greater than that of spring thaw period due to the mean soil temperature of the different specified periods. The nongrowing season $N_2O$ emissions ranged from $0.89$ to $1.44$ kg ha$^{-1}$, which contributed to $41.96$–$53.73\%$ of the annual budget, accounting for almost half of the annual emissions in the permafrost region. The driving factors of $N_2O$ emissions were different among during the study period, growing season, and nongrowing season. The $N_2O$ emissions from total two-year observation period and nongrowing season were mainly affected by soil temperature, while the $N_2O$ emissions from growing season were controlled by soil temperature, water table level, and their interactions. In conclusion, nongrowing season $N_2O$



emissions is an important component of annual emissions and cannot be ignored in
the permafrost region.

**1 Introduction**

Permafrost regions cover approximately 25% of terrestrial land and store large
amounts of nitrogen stocks (31–102 Pg) in soils (Harden et al., 2012). With climate
warming, permafrost regions are thawing and degrading globally (IPCC, 2013). Large
amounts of soil nitrogen have been released to the atmosphere from the permafrost
region. Nitrous oxide ($N_2O$) is a major component of N exchanged between terrestrial
ecosystems and atmosphere in the permafrost region and feedback to climate warming.
$N_2O$ is the third most important greenhouse gas with 265 times the global warming
potential of $CO_2$ and 9 times that of $CH_4$, which contributes 6% to global climate
warming (IPCC, 2013). Soil biological processes, which release approximately 60%
of total natural $N_2O$ emissions, are the largest source of $N_2O$ emissions to the
atmosphere (IPCC, 2013). Permafrost regions were considered to release negligible
amounts of $N_2O$ emission because of the limited mineral N content. Recently, "hot
spots" for $N_2O$ emissions from permafrost regions were found in the subarctic
(Marushchak et al., 2011;Repo et al., 2009). The rates of $N_2O$ emissions from bare
peatland could reach 31–31.4 mg $m^{-2}$ $day^{-1}$, which are as high as $N_2O$ emissions from
tropical soil (Marushchak et al., 2011;Repo et al., 2009;Castaldi et al., 2013). The
cumulative $N_2O$ emissions range from 0.9 to 1.4 g $m^{-2}$ during the growing season,
indicating that the permafrost region is also an important source of $N_2O$ emissions



(Repo et al., 2009). In the past, research on $N_2O$ emissions from permafrost regions
were mainly focused on the growing season (Repo et al., 2009;Gao et al., 2019b;Chen
et al., 2017). However, in the permafrost region, $N_2O$ emissions from the nongrowing
season are unclear.

N$_2$O emissions have been widely researched during the nongrowing season in
different ecosystems (Maljanen et al., 2010;Merbold et al., 2013;Furon et al., 2008). A
significant release of $N_2O$ emissions have been observed during the nongrowing
season, particularly during the spring thaw period. During the nongrowing season, the
rates of $N_2O$ emission could be more than 230 g N ha$^{-1}$ d$^{-1}$ (Glenn et al., 2012;Flesch
et al., 2018;Chantigny et al., 2017) and the cumulative $N_2O$ emissions released can be
as high as 40 kg ha$^{-1}$ in agricultural soil (Dunmola et al., 2010). The importance of
nongrowing season $N_2O$ emissions to the annual budget, which contributed more than
50% of the annual values, have been shown in different ecosystems (Fu et al.,
2018;Virkajärvi et al., 2010;Yanai et al., 2011). Scientists have focused primarily on
$N_2O$ emissions during the nongrowing season in the agricultural (Furon et al.,
2008;Dietzel et al., 2011), grassland (Virkajärvi et al., 2010;Merbold et al., 2013),
forest (Maljanen et al., 2010), wetland (Hao et al., 2006), and tundra ecosystems
(Brooks et al., 1997). Nongrowing season $N_2O$ emissions are an essential component
of global N cycling. Permafrost regions, which are characterized by cold temperatures,
are mainly distributed in high-latitude and high-altitude areas, and are extremely
sensitive to climatic warming. The nongrowing season lasts for more than half of the





year in the permafrost region. Determining nongrowing season $N_2O$ emissions is
important for accurately evaluating annual $N_2O$ emission from permafrost regions.
However, $N_2O$ emissions from permafrost regions still remain uncertain during the
nongrowing season.

Daxing'an Mountains, located in Heilongjiang province of Northeast China, are
a unique high latitude and the second largest permafrost region in China. Under the
threat of global warming, the permafrost region in the Daxing'an Mountains has been
significantly degrading and thawing (Jin et al., 2007). The area of the permafrost
region has decreased by 35%, leading to the deepening of the active layer, thinning of
the permafrost layer, and increasing ground temperatures, which changes the N cycle
(Jin et al., 2007). The previous in-situ $N_2O$ measurements from permafrost region of
the Daxing'an Mountains have primarily been reported during the growing season or
the spring thaw period (Gao et al., 2019b;Cui et al., 2018;Gao et al., 2019a). In the
context of global climate warming, $N_2O$ emission during the nongrowing season are
unclear in the permafrost region of the Daxing'an Mountains.

The typical vegetation in the permafrost region of the Daxing'an Mountains is
cool-temperate coniferous forest dominated by *Larix gmelinii*, forming the southern
boundary of the boreal forest. In this study, in-situ $N_2O$ emission were measured from
the permafrost region at three forest sites in the Daxing'an Mountains for two full
years. The objectives of this study were to: (i) characterize the nongrowing season





N$_2$O emissions from continuous permafrost regions; (ii) evaluate the contributions
from the nongrowing season, particularly the spring thaw period, to annual N$_2$O
emissions; and (iii) investigations the key regulatory factors on N$_2$O emissions.
Observation of the nongrowing season N$_2$O emissions from permafrost regions
provides insight into regional climate warming and the impact of the permafrost
region on global climate change.

**2 Materials and Methods**
**2.1 Site description**
The experimental site was located on the continuous permafrost region in the
Heilongjiang Mohe Forest Ecosystem Research Station at the Daxing'an Mountains,
Northeast China (122°06′–122°27′E, 53°17′–53°30′N; 290–740 m elevation). The
study region has a typical cold temperate continental climate with a long cold winter
and short hot summer. Air temperature ranges from −52.3 to 36.6 °C with a mean
annual temperature of −4.9 °C. Mean annual precipitation is 430–550 mm, 60% of
which falls as rain primarily in the summer. Snow accumulation is 20–40 cm and
covers the land for more than half of the year (from October to April). The soil at the
study site is primarily brown forest soil, interspersed with meadow soil and marsh
soil.

The typical vegetation in this permafrost region is a temperate coniferous forest
with *L. gmelinii* as the dominant species. Other overstory species include *Betula*



*platyphylla*, *Pinus sylvestris* var. *mongolica*. Shrub species include *Ledum palustre*
var. *dilatatum*, *B. fruticose*, *Vaccinium uliginosum*, *V. vitis-idaea, Rhododendron*
*dauricum*, and *Alnus sibirica*. Herbaceous species include *Carex appendiculata*, *C.*
*schmidtii*, *Eriophorum vaginatum*, *Rubus clivicola*, and *Sanguisorba officinalis*.
According the water table level from low to high, three types of typical swamp forests
located in the permafrost region were studied: *L. gmelinii - Ledum palustre* var.
*dilatatum* swamp forest (*LL*), *L. gmelinii - Carex appendiculata* swamp forest (*LC*),
and *Betula fruticose* swamp forest (*B*). The soil physicochemical properties at the
three swamp forests are shown in table 1.

















**Table 1.**
Soil properties at the three swamp forest sites in the permafrost region of Daxing'an
Mountains, Northeast China (mean ± SD).

| Environmental factor | *LL* site | *LC* site | *B* site |
|---|---|---|---|
| WTL | −13.19 ± 9.73b | −4.51 ± .51a | 0.26 ± 5.67a |
| $SM_{0-10}$ | 117.30 ± 14.92c | 174.20 ± 14.58a | 162.31 ± 16.14b |
| $SM_{10-20}$ | 49.54 ± 8.28b | 115.86 ± 10.98a | 115.91 ± 9.13a |
| $pH_{0-10}$ | 4.77 ± 0.16c | 4.99 ± 0.08a | 4.89 ± 0.11b |
| $pH_{10-20}$ | 4.93 ± 0.18c | 5.09 ± 0.07a | 4.99 ± 0.11b |
| $NH_4^+\text{-}N_{0-10}$ | 5.49 ± 2.15a | 5.98 ± 3.03a | 4.92 ± 2.65a |
| $NH_4^+\text{-}N_{10-20}$ | 3.05 ± 1.57a | 3.87 ± 1.94a | 3.43 ± 1.88a |
| $NO_3^-\text{-}N_{0-10}$ | 1.71 ± 0.73a | 1.81 ± 1.02a | 1.58 ± 0.63a |
| $NO_3^-\text{-}N_{10-20}$ | 1.29 ± 0.57ab | 1.44 ± 1.02a | 1.02 ± 0.42b |
| $TOC_{0-10}$ | 39.95 ± 6.91a | 42.01 ± 4.43a | 35.57 ± 5.22b |
| $TOC_{10-20}$ | 15.62 ± 3.95b | 18.25 ± 2.71a | 16.62 ± 2.1ab |
| $TN_{0-10}$ | 2.19 ± 0.37b | 3.78 ± 0.51a | 1.97 ± 0.69b |
| $TN_{10-20}$ | 0.83 ± 0.15b | 1.03 ± 0.21a | 0.91 ± 0.13b |
| $C/N_{0-10}$ | 17.94 ± 4.17a | 11.27 ± 144b | 17.08 ± 3.55a |
| $C/N_{10-20}$ | 19.26 ± 5.65a | 18.29 ± 4.24a | 18.57 ± 3.82a |

WTL, water table level; $SM_{0-10}$, soil moisture at 0–10 cm; $SM_{10-20}$, soil moisture
at 10–20 cm; $pH_{0-10}$, pH at 0–10 cm; $pH_{10-20}$, pH at 10–20 cm; $NH_4^+\text{-}N_{0-10}$,
ammonium nitrogen at 0–10 cm; $NH_4^+\text{-}N_{10-20}$, ammonium nitrogen at 10–20 cm;





$NO_3^-$-$N_{0–10}$, nitrate nitrogen at 0–10 cm; $NO_3^-$-$N_{10–20}$, nitrate nitrogen at 10–20 cm;
$TOC_{0–10}$, total organic carbon at 0–10 cm; $TOC_{10–20}$, total organic carbon at 10–20 cm;
$TN_{0–10}$, total nitrogen at 0–10 cm; $TN_{10–20}$, total nitrogen at 10–20 cm; $C/N_{0–10}$,
carbon-to-nitrogen ratio at 0–10 cm; $C/N_{10–20}$, carbon-to-nitrogen ratio at 10–20 cm.

**2.2 N₂O emission measurements**
The field experiment was conducted from June 2016 to June 2018. Three $20 \times 20$
m plots were permanently established at each *LL*, *LC*, and *B* site, respectively. N₂O
emissions were measured with the static opaque chamber technique (Hutchinson et al.,
2000). The polypropylene chamber collar with a water-filled channel (50 cm × 50 cm
× 20 cm height) was randomly inserted 20 cm into the soil. Gas samples were
measured from 9:00 am to 11:00 am, the hours that were most representative of the
daily mean N₂O emissions (Alves et al., 2012). During each N₂O measurement period,
chambers (50 cm × 50 cm × 50 cm height) were sealed by filling the collars with
water and used to collect N₂O from the soil. Four-chamber headspace air samples
were taken using a 50-mL plastic syringe at 0, 15, 30, and 45 min after chamber
closure (Liu et al., 2019). Samples were injected into pre-evacuated 100 mL gas
sampling bags (Delin Gas Packing Co., Dalian, China) for subsequent laboratory
analysis. The air temperature inside the chamber was recorded when gas samples were
being retrieved. Gas samples were taken twice per month during the growing season
from June to September, monthly during the winter from October to December, and
every three to ten days during the spring thaw period from March to May (45





sampling events in total).

The gas $N_2O$ concentration was analyzed with a gas chromatograph coupled with
an electron capture detector (ECD) (Shimadzu GC2010, Shimadzu Analytical and
Measuring Instruments Division, Kyoto, Japan). Compressed air containing 0.378
ppm $N_2O$ was used for calibration. $N_2$ was used as the carrier gas with a flow rate of
20 mL $min^{-1}$. The $N_2O$ was separated using a 1-m stainless steel column with an inner
diameter of 2 mm from Porapak Q (80/100 mesh), and was detected by an ECD. The
temperature for gas separation was maintained at 70 °C and the detector was set at
250 °C.

**2.3 Measurements of meteorological and soil physiochemical properties**
Soil temperature (ST) at 5, 10, and 15 cm deep was monitored at each collar
using a portable digital thermometer with a thermocouple probe (JM-624, Jinming
Corp., Tianjin, China). During the growing season, the water table level (WTL) was
measured near the chamber in each plot using a ruler (Dobbie and Smith, 2006). In
the nongrowing season, soil moisture was determined by the oven-drying mothed.

Using a 3.8 cm diameter stainless-steel sampling probe, soil samples were taken
from the upper 0–10 cm and lower 10–20 cm soil layers close to each collar. Fine
roots and visible organic debris were removed by passing the soil samples through a
2-mm sieve. Then, samples were stored in an insulated box (Esky) and stored at 4 °C





for subsequent chemical analysis.

Soil moisture contents were determined by drying at 105 °C for 48 h followed by
calculating the weight loss. Soil samples were air dried and sieved to <2 mm
aggregate size and used to measure the soil pH. Soil pH was measured in a 2:5
air-dried soil: deionized water mixture using an InoLab pH meter (WTW InoLab pH
730, Weilheim, Germany). Soil mineral N, ammonium ($NH_4^+$-N), and nitrate ($NO_3^-$-N)
content were determined on 10 g samples of fresh soil based on a 1 mol/L KCl
solution extraction procedure. The extracts were filtered through a 0.45 µm
pore-diameter syringe filter, and then soil mineral N was analyzed using a Lachat
flow-injection auto-analyzer (Seal Analytical AA3, Norderstedt, Germany). For the
analyses of total N and C, sub-samples were further ground to a fine powder (<0.15
mm). Total N concentration was measured on aliquots of 1.0 g of soil using
semi-micro-Kjeldahl method. The TOC content was determined by oxidation at
170 °C, with potassium dichromate in the presence of sulfuric acid. The excess
potassium dichromate was titrated with a solution of Mohr's salt.

**2.4 Statistical analysis**
The $N_2O$ emissions were calculated as reported by (Hou et al., 2012). A positive
regression indicates the emission from soil to the atmosphere. A negative regression
indicates a net uptake by the soil from the atmosphere. Cumulative $N_2O$ emissions
were linearly and sequentially accumulated from the emissions between every two





adjacent intervals of the measurements following the procedure described by Ding et
al. (2007).

The calendar year was divided into three seasons: winter, spring thaw period, and
growing season. The winter was defined as the period during which the daily mean air
temperature remained below 0 °C for at least five consecutive days. The spring thaw
period was defined as the period when the daily maximum air temperature exceeded
0 °C and ended at soil thawing to a depth of 20 cm. The growing season was defined
as the period that lasts from the end of the spring thaw period to the beginning of
winter. We designated the winter and spring thaw period as the nongrowing season.

A one-way analysis of variance was used to test the difference of $N_2O$ emissions
and environment factors in the three swamp forest sites. A T-test was used to identify
the differences in $N_2O$ emissions between 2016/2017 and 2017/2018. The correlations
between the $N_2O$ emissions and environmental factors were tested using Pearson's
correlation analysis. A linear correlation analysis and multivariate regression analysis
were conducted to create explanatory models using the same variables as those used
for describing the temporal variation of $N_2O$ emissions. R software (Version 3.4.1,
https://www.r-project.org/) was used for the statistical analyses. Significance was
analyzed using Fisher's least significant difference (LSD) test at a probability level of
95% ($P < 0.05$). All figures were drawn using OriginPro 2018 software (OriginLab
Corp., Northampton, MA, U.S.A.).






## 3 Results

### 3.1 Temporal variation of N₂O emissions

During the two-year observation period, there was significant temporal variation in N₂O emissions in the permafrost region of the Daxing'an Mountains (Fig. 1a). However, the temporal pattern of N₂O emissions were different in the three swamp forests. The N₂O emissions ranged from −17.40 to 74.16, −35.75 to 64.73, and −26.47 to 79.25 $\mu g \cdot m^{-2} \cdot h^{-1}$ in the *LL*, *LC*, and *B* sites, respectively (Fig. 1a). The highest N₂O emissions occurred in different periods in three swamp forests, namely, they occurred in the beginning of the spring thaw period in the *LL* site and at the end of growing season in the *LC* and *B* sites. The lowest N₂O emissions from the three swamp forest sites were all observed in the spring thaw period. Negative emissions mainly occurred during the winter and spring thaw period. The N₂O emissions during the nongrowing season mainly ranged from −17.40 to 74.16, −35.75 to 60.76, and −26.47 to 71.82 $\mu g \cdot m^{-2} \cdot h^{-1}$ in the *LL*, *LC*, and *B* sites, respectively.

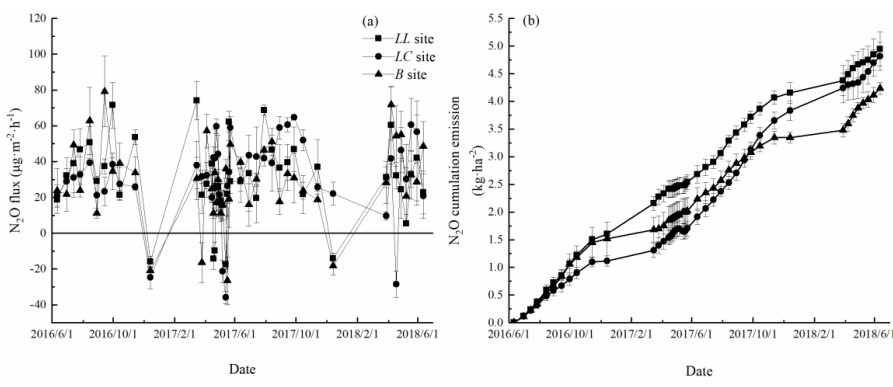


**Figure 1.** N₂O emissions (A) and cumulative N₂O emissions (B) from three types of



swamp forests in the permafrost region of the Daxing'an Mountains, Northeast China

The annual mean $N_2O$ emissions ranged from 27.80 to 32.51, 26.10 to 37.89, and
25.86 to 33.16 $\mu g \cdot m^{-2} \cdot h^{-1}$ in the *LL*, *LC*, and *B* sites, respectively (Table 2). The mean
$N_2O$ emissions from the growing season typically higher than that of winter and the
spring thaw period in the three swamp forest sites. In 2016/2017, the mean $N_2O$
emissions were all highest in the growing season and lowest in the winter. In contrast,
during 2017/2018, the mean $N_2O$ emissions were lowest during the spring thaw
period in the *LC* site and highest during the spring thaw period in the *B* site. For the
different types of swamp forests, the mean $N_2O$ emissions from the *LC* site were
significantly higher than the mean $N_2O$ emissions in the *B* site in the 2017/2018
growing season. There was no significant difference in $N_2O$ emissions during the
winter, spring thaw period, and annually in the three swamp forests. The $N_2O$
emissions from different periods were generally not significantly different between
the two years. Differences in $N_2O$ emissions were found during the growing season
and spring thaw period. The mean $N_2O$ emissions during the growing season from the
*LC* site and the $N_2O$ emissions during the spring thaw period from the *B* site were
both significantly higher in 2017/2018 than 2016/2017.





**Table 2.**

Summary of N₂O emissions during specified periods from the three swamp forest sites in the permafrost region of the Daxing'an Mountains, Northeast China

| Specified period | Duration (Days) | | Mean N$_2$O emissions ($\mu g \cdot m^{-2} \cdot h^{-1}$) | | | | | |
| --- | --- | --- | --- | --- | --- | --- | --- | --- |
| | | | 2016/2017 | | | 2017/2018 | | |
| | 2016/2017 | 2017/2018 | LL site | LC site | B site | LL site | LC site | B site |
| Growing season | 134 | 129 | 40.42 ± 15.97Aa | 29.54 ± 7.30Ab | 38.35 ± 23.42Aa | 40.09 ± 14.68ABa | 47.73 ± 12.27Aa | 33.17 ± 12.48Ba |
| Nongrowing season | | | | | | | | |
| Winter | 155 | 161 | 19.80 ± 34.74Aa | 9.62 ± 29.69Aa | 17.41 ± 33.14Aa | 14.95 ± 26.23Aa | 33.31 ± 16.25Aa | 8.13 ± 22.89Aa |
| Spring thaw period | 77 | 75 | 22.49 ± 24.90Aa | 27.56 ± 26.06Aa | 20.89 ± 21.73Ab | 32.74 ± 16.69Aa | 31.04 ± 31.31Aa | 41.66 ± 18.78Aa |
| Annual | 366 | 365 | 27.80 ± 24.36Aa | 26.10 ± 22.44Aa | 25.86 ± 24.06Aa | 32.51 ± 18.31Aa | 37.89 ± 22.26Aa | 33.16 ± 19.55Aa |

The different capital letters indicate that the N₂O emissions were significantly different among the types of swamp forest; different lowercase letters indicate that the N₂O emissions were significantly different between the two years.





**3.2 Seasonal contribution of N₂O emissions to the annual budget**

Cumulative $N_2O$ emissions primarily increased during the study period (Fig. 1b). The annual $N_2O$ emissions ranged from 2.27 to 2.68, 1.92 to 2.90, and 2.00 to 2.24 kg ha$^{-1}$ yr$^{-1}$ in the *LL*, *LC*, and *B* sites, respectively (Table 3). The cumulative $N_2O$ emissions during the growing season ranged from 1.02 to 1.46 kg ha$^{-1}$, which contributed to 46.27 to 58.04% to the annual emissions. The cumulative $N_2O$ emissions from the growing season were higher than the cumulative $N_2O$ emissions during the winter and spring thaw periods, and were 1.2 to 3.2 times greater than that of the winter and 1.5 to 3.7 times greater than that of the spring thaw period. The cumulative $N_2O$ emissions during the nongrowing season were mainly lower than during the growing season, which contributed to 41.96–53.73% to annual emissions. The cumulative $N_2O$ emissions during the spring thaw period ranged from 0.35 to 0.66 kg ha$^{-1}$, contributing to 15.63 to 33.00% to the annual emissions.




**Table 3.**

The cumulative $N_2O$ emissions and its contribution to annual $N_2O$ emissions from the three swamp forest sites in the permafrost region of

Daxing'an Mountains, Northeast China.

| Forest types | Year | Cumulative $N_2O$ emissions (kg ha$^{-1}$) | | | | | Contribution to annual $N_2O$ emissions (%) | | | |
|---|---|---|---|---|---|---|---|---|---|---|
| | | Annual | GS | NGS | Winter | STP | GS | NGS | Winter | STP |
| LL site | 2016/2017 | 2.68 ± 0.15 | 1.24 ± 0.06 | 1.44 ± 0.21 | 1.00 ± 0.20 | 0.44 ± 0.04 | 46.27 | 53.73 | 37.31 | 16.42 |
| | 2017/2018 | 2.27 ± 0.16 | 1.21 ± 0.07 | 1.06 ± 0.18 | 0.56 ± 0.16 | 0.50 ± 0.03 | 53.30 | 46.70 | 24.67 | 22.03 |
| LC site | 2016/2017 | 1.92 ± 0.14 | 1.03 ± 0.13 | 0.89 ± 0.13 | 0.48 ± 0.11 | 0.41 ± 0.02 | 53.65 | 46.35 | 25.00 | 21.35 |
| | 2017/2018 | 2.90 ± 0.11 | 1.46 ± 0.03 | 1.44 ± 0.10 | 0.93 ± 0.12 | 0.51 ± 0.03 | 50.34 | 49.66 | 32.07 | 17.59 |
| B site | 2016/2017 | 2.24 ± 0.21 | 1.30 ± 0.15 | 0.94 ± 0.08 | 0.58 ± 0.07 | 0.35 ± 0.01 | 58.04 | 41.96 | 25.89 | 15.63 |
| | 2017/2018 | 2.00 ± 0.22 | 1.02 ± 0.11 | 0.98 ± 0.22 | 0.32 ± 0.20 | 0.66 ± 0.08 | 51.00 | 49.00 | 16.00 | 33.00 |

GS: growing season; NGS: nongrowing season; STP: spring thaw period.



### 3.3 Temporal control of soil N$_2$O emissions

The relationship between N$_2$O emissions and environmental factors during the different specified periods are shown in Table 4. During the entire two-year observation period, the N$_2$O emission were all significantly positively correlated with soil temperature at 5, 10, and 15 cm in the three swamp forest sites. Except for the soil temperature, the N$_2$O emissions from the *LL* site were also positively correlated with air temperature ($P<0.05$) and C/N$_{0-10}$ ($P<0.05$) and negatively correlated with pH$_{0-10}$ ($P<0.01$), pH$_{10-20}$ ($P<0.05$), NO$_3^-$-N$_{0-10}$ ($P<0.05$), NO$_3^-$-N$_{10-20}$ ($P<0.05$), and TN$_{0-10}$ ($P<0.05$). The N$_2$O emissions from the *LC* site were significantly positively correlated with TOC$_{0-10}$ ($P<0.01$) and TN$_{10-20}$ ($P<0.01$). The N$_2$O emissions from the *B* site were significantly positively correlated with air temperature ($P<0.001$), NH$_4^+$-N$_{0-10}$ ($P<0.05$), and NH$_4^+$-N$_{10-20}$ ($P<0.05$). Similar to the entire period, the N$_2$O emissions from the nongrowing season were mainly significantly positively correlated with soil temperature in the *LC* and *B* sites and weakly positively correlated with soil temperature in the *LL* sites. The N$_2$O emissions from the *LL* site were also significantly negatively correlated with pH$_{0-10}$ ($P<0.05$) and TN$_{0-10}$ ($P<0.001$); the N$_2$O emissions from the *LC* site were significantly positively correlated with TN$_{0-10}$ ($P<0.05$). The N$_2$O emissions from the *B* site were significantly positively correlated with air temperature ($P<0.05$), NH$_4^+$-N$_{0-10}$ ($P<0.01$), NH$_4^+$-N$_{10-20}$ ($P<0.01$), and NO$_3^-$-N$_{0-10}$ ($P<0.05$). For the growing season, the impact of environmental factors on N$_2$O emissions were complicated. The N$_2$O emissions were significantly positively correlated with $T_{15}$ ($P<0.05$) and negatively correlated with NO$_3^-$-N$_{0-10}$ ($P<0.05$) and

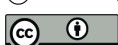



$NO_3^-$-$N_{10-20}$ (P<0.01) in the *LL* site. The $N_2O$ emissions from *LC* site were
significantly influenced by air temperature, water table level, $NO_3^-$-N, TOC, TN, and
C/N ratio. The $N_2O$ emissions from the *B* site were only significantly positively
correlated with water table level (P<0.05).



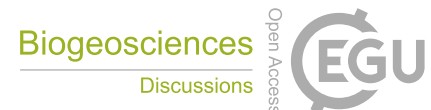

**Table 4.**
The relationship between $N_2O$ emissions and environmental factors from three swamp forest sites in the permafrost region of the Daxing'an
Mountains, Northeast China.

| Environmental factor | Two full-year | | | Growing season | | | Nongrowing season | | |
|---|---|---|---|---|---|---|---|---|---|
| | LL site | LC site | B site | LL site | LC site | B site | LL site | LC site | B site |
| $Ta$ | 0.17* | 0.14 | 0.30*** | −0.26[+] | −0.50*** | 0.15 | 0.11 | 0.16 | 0.26* |
| $T_5$ | 0.33*** | 0.26** | 0.35*** | 0.07 | −0.13 | 0.19 | 0.17 | 0.25* | 0.29** |
| $T_{10}$ | 0.37*** | 0.28** | 0.36*** | 0.20 | −0.03 | 0.21 | 0.20[+] | 0.26* | 0.31** |
| $T_{15}$ | 0.39*** | 0.30*** | 0.34*** | 0.30* | 0.06 | 0.14 | 0.21[+] | 0.32** | 0.32** |
| WTL | | | | −0.13 | −0.29* | 0.29* | | | |
| $SM_{0-10}$ | | | | | | | −0.08 | −0.02 | 0.03 |
| $SM_{10-20}$ | | | | | | | −0.10 | 0.08 | −0.03 |
| $pH_{0-10}$ | −0.24** | 0.07 | −0.18[+] | −0.01 | 0.08 | −0.20 | −0.25* | 0.08 | −0.15 |



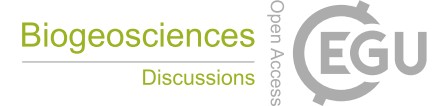

| | | | | | | | | | |
|---|---|---|---|---|---|---|---|---|---|
| $pH_{10-20}$ | −0.21* | 0.15 | −0.12 | −0.05 | 0.14 | 0.05 | −0.17 | 0.22+ | −0.19 |
| $NH_4^+\text{-}N_{0-10}$ | 0.08 | 0.05 | 0.20* | 0.21 | 0.19 | −0.06 | 0.07 | −0.11 | 0.32** |
| $NH_4^+\text{-}N_{10-20}$ | 0.05 | 0.01 | 0.24* | −0.08 | −0.23 | 0.14 | 0.21+ | 0.15 | 0.34** |
| $NO_3^-\text{-}N_{0-10}$ | −0.24* | 0.05 | 0.01 | −0.32* | 0.28* | −0.16 | −0.08 | 0.09 | 0.29* |
| $NO_3^-\text{-}N_{10-20}$ | −0.22* | 0.09 | −0.11 | −0.36** | 0.36* | −0.02 | 0.01 | 0.11 | 0.03 |
| $TOC_{0-10}$ | 0.13 | 0.26** | −0.14 | 0.23 | 0.48*** | −0.15 | −0.16 | 0.01 | −0.05 |
| $TOC_{10-20}$ | 0.08 | 0.10 | −0.06 | 0.03 | −0.02 | −0.01 | −0.14 | 0.11 | −0.09 |
| $TN_{0-10}$ | −0.21* | 0.06 | −0.03 | 0.08 | −0.35* | −0.04 | −0.43*** | 0.14 | 0.03 |
| $TN_{10-20}$ | 0.01 | 0.26** | −0.16+ | 0.19 | 0.65*** | −0.16 | −0.06 | 0.26* | −0.15 |
| $C/N_{0-10}$ | 0.21* | 0.15 | −0.08 | 0.13 | 0.53*** | −0.05 | 0.10 | −0.17 | −0.14 |
| $C/N_{10-20}$ | 0.06 | −0.14 | 0.04 | −0.18 | −0.54*** | 0.04 | −0.13 | −0.13 | 0.05 |

+: indicates significant effects at $P < 0.1$; *: indicates significant effects at $P < 0.05$; **: indicates significant effects at $P < 0.01$; ***:
indicates significant effects at $P < 0.001$.





The Pearson correlation analysis showed that the soil temperature was the key
environmental factor controlling the $N_2O$ emissions during the entire observation
period and the nongrowing season. During the entire observation period, soil
temperature at 5, 10, and 15 cm could explain 10.39 to 14.48, 6.07 to 8.34, and 10.66
to 12.02% of the temporal variation of $N_2O$ emissions in the *LL*, *LC*, and *B* sites,
respectively. During the nongrowing season, $N_2O$ emissions from the *LL* site were
weakly positively correlated with soil temperature, explaining 1.73 to 3.27% of $N_2O$
emissions. The soil temperature could explain 5.02 to 9.54% and 7.51 to 9.36% of the
$N_2O$ fluctuation in the *LC* and *BC* sites, which were lower than the entire observation
period.



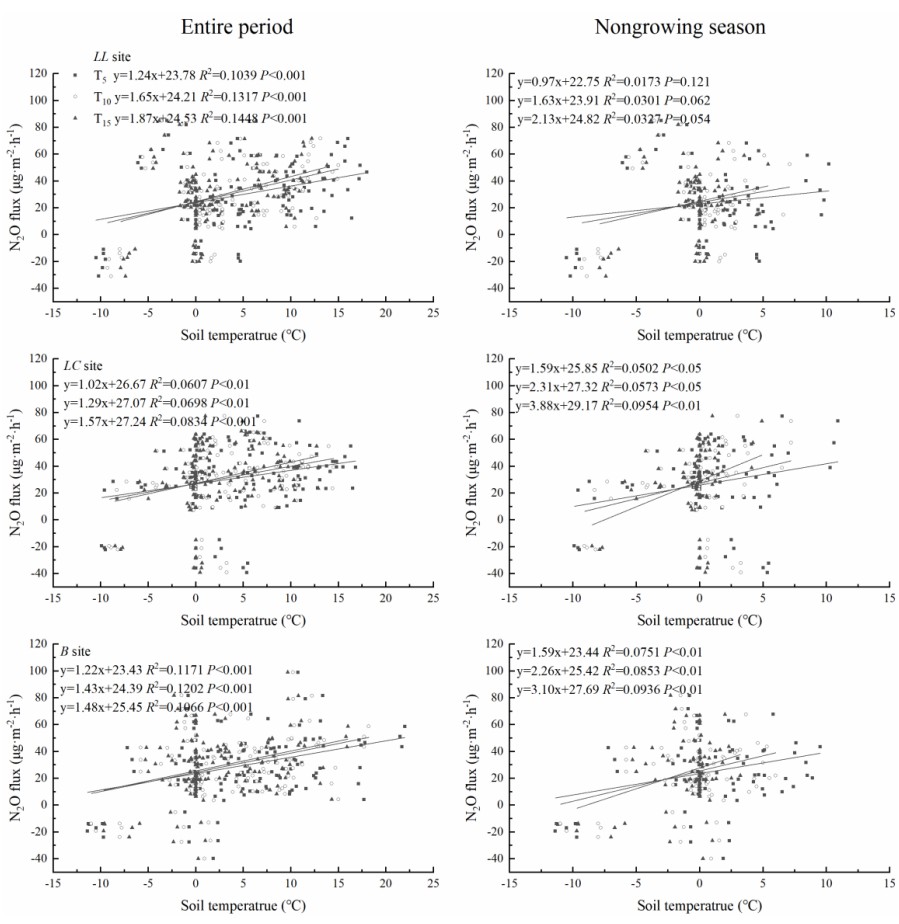


**Figure 2.** The linear models between N$_2$O emissions and soil temperature during the

entire observation period and nongrowing season in the three swamp forest sites in the

permafrost region of the Daxing'an Mountains, Northeast China.


During the growing season, multivariate regression analyses showed that the

N$_2$O emissions were affected by soil temperature, water table level, and their
interactions (Table 5). Soil temperature, water table level, and their interactions could
explain 26.35, 19.46, and 12.36% of the temporal variation of N$_2$O emissions in the
three swamp forest sites, respectively.






**Table 5.**
Models of $N_2O$ emissions during the growing season against soil temperature and
water table level for the three swamp forest sites in the permafrost region of the
Daxing'an Mountains, northeast China.

| Forest type | a | b | c | d | $R^2$ | P |
|---|---|---|---|---|---|---|
| *LL* site | −11.55 | 4.03 | −3.19 | 0.25 | 0.2635 | <0.001 |
| *LC* site | 22.44 | 0.94 | −3.99 | 0.28 | 0.1946 | <0.01 |
| *B* site | 27.57 | 0.77 | −1.50 | 0.21 | 0.1236 | <0.05 |

The regression models are: $y = a + b \times T_5 + c \times WTL + d \times T_5 \times WTL$, where a, b, c, and d
are the regression coefficients.

**4 Discussion**
**4.1 Soil temperature controls the mean $N_2O$ emissions from the different periods**
In previous studies, the $N_2O$ emissions from the permafrost region primarily
focused on emissions during the growing season (Gil et al., 2017;Lamb et al.,
2011;Voigt et al., 2017). Permafrost regions are mainly distributed in high-altitude and
high-latitude zones. It was difficult to measure $N_2O$ emissions during the nongrowing
season in the cold climate conditions of the permafrost region. Publications on
nongrowing season $N_2O$ emissions are scarce and the difference of mean $N_2O$
emissions among the winter, spring thaw period, and growing season are unknown in
the permafrost region.






380 The nongrowing season $N_2O$ emission ranged from $-35.75$ to $74.16\ \mu g\cdot m^{-2}\cdot h^{-1}$

381 in the permafrost region of the Daxing'an Mountains, northeast China. The results

382 were similar to the rate of annual $N_2O$ emission ($-35.75$ to $79.25\ \mu g\cdot m^{-2}\cdot h^{-1}$) in the

383 permafrost region of the Daxing'an Mountains and within the range of $N_2O$ emissions

384 reported in permafrost ecosystems ($-35.75$ to $2662\ \mu g\cdot m^{-2}\cdot h^{-1}$) (Gao et al., 2019a;Mu

385 et al., 2017). The $N_2O$ emissions confirmed our previous findings that the $N_2O$

386 emissions from the Daxing'an Mountains ranged within the intermediate range for

387 permafrost ecosystems (Gao et al., 2019b).

389 The annual $N_2O$ emissions showed significant temporal variations in grasslands

390 (Du et al., 2006). There were significant differences in the mean $N_2O$ emissions in the

391 spring, summer, autumn, and winter in grasslands, whereas the temporal pattern over

392 the course of the five-year study (Du et al., 2006). The $N_2O$ was taken up during the

393 freezing period and emitted during the thawing period and growing season in marshes,

394 indicating that the emissions were different among the three specified periods (Hao et

395 al., 2006). These trends were also observed in the permafrost region. In the "hot spots"

396 of $N_2O$ emission from permafrost region, high $N_2O$ emissions were observed during

397 the nongrowing season in the bare peatland, which contributed 20–69% to the annual

398 emissions from the bare peatland (Marushchak et al., 2011). In the vegetated

399 permafrost ecosystem, the $N_2O$ emissions during the nongrowing season, growing

400 season, and annually were mainly negligible (Marushchak et al., 2011). The $N_2O$



emissions from the spring thaw period and nongrowing season were lower than that of
growing season in the permafrost region of the Daxing'an Mountains (Chen et al.,
2017;Gao et al., 2019a;Wu et al., 2019). However, the drivers of $N_2O$ emissions
between the nongrowing season and growing season were not clear in the permafrost
region. Our results showed that the $N_2O$ emissions were the highest during the
growing season and lowest during the nongrowing season. The mean $N_2O$ emissions
from the growing season were significantly higher than the emissions from the winter
in the *LL* and *B* sites, whereas the $N_2O$ emissions during the spring thaw period was
not significantly different from growing season and winter in the three swamp forests
(Fig. 3). The $N_2O$ emissions from the growing season were 1.75–2.86 times greater
than the winter emissions and 1.31–1.53 times greater than during the spring thaw
period in the three forest types.

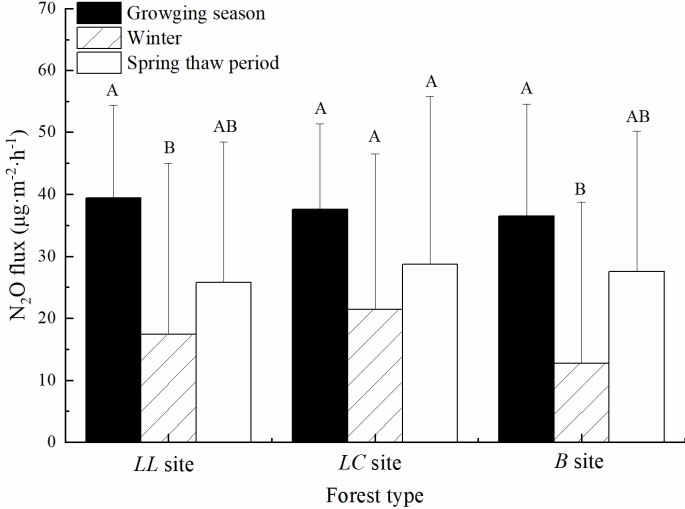


**Figure 3.** The difference of $N_2O$ emissions among the growing season, winter, and
spring thaw period in the permafrost region of the Daxing'an Mountains, Northeast



China.

We found that the mean $N_2O$ emissions of the three specified periods were

significantly positively correlated with soil temperature at 5, 10, and 15 cm, which
could explain 91.36–94.07, 91.97–95.92, and 81.71–92.85% of the temporal variation
of mean $N_2O$ emissions in the *LL*, *LC*, and *B* sites, respectively (Fig. 4). In a
laboratory experiment, the $N_2O$ emissions were strongly dependent on soil
temperatures above zero (Oquist et al., 2004). The net $N_2O$ production rates at −4 °C
equaled those observed at +10 to +15 °C in the boreal forest soil (Oquist et al., 2004).
However, the field soil temperature in the winter in the Daxing'an Mountains was
significantly lower than the simulated cold temperature in the laboratory, meaning that
the winter $N_2O$ emissions may be lower than $N_2O$ emissions during the growing
season (Oquist et al., 2004). In the nongrowing season, soil moisture was consistently
saturated in the 0–10 cm soil layer of three swamp forests, which implies that $N_2O$
production occurred predominantly due to denitrification. The denitrification rates
showed similar temporal variations of $N_2O$ emissions in agricultural soil in the winter
(Tatti et al., 2014). The copy numbers of denitrifier genes (*nirS* and *nirK*) remained
stable from November to January and increased in March and April, indicating that
$N_2O$ emissions during the spring thaw period were higher than during the winter (Tatti
et al., 2014). During the growing season, $N_2O$ emissions were significantly positively
correlated with soil temperature in the permafrost region (Marushchak et al., 2011;Cui
et al., 2018;Chen et al., 2017). Thus, the soil temperature controlled the mean $N_2O$





emissions during the winter, spring thaw period, and growing season. Soil
temperature was the key environmental factor determining the temporal variation of
$N_2O$ emissions in the permafrost region of the Daxing'an Mountains.

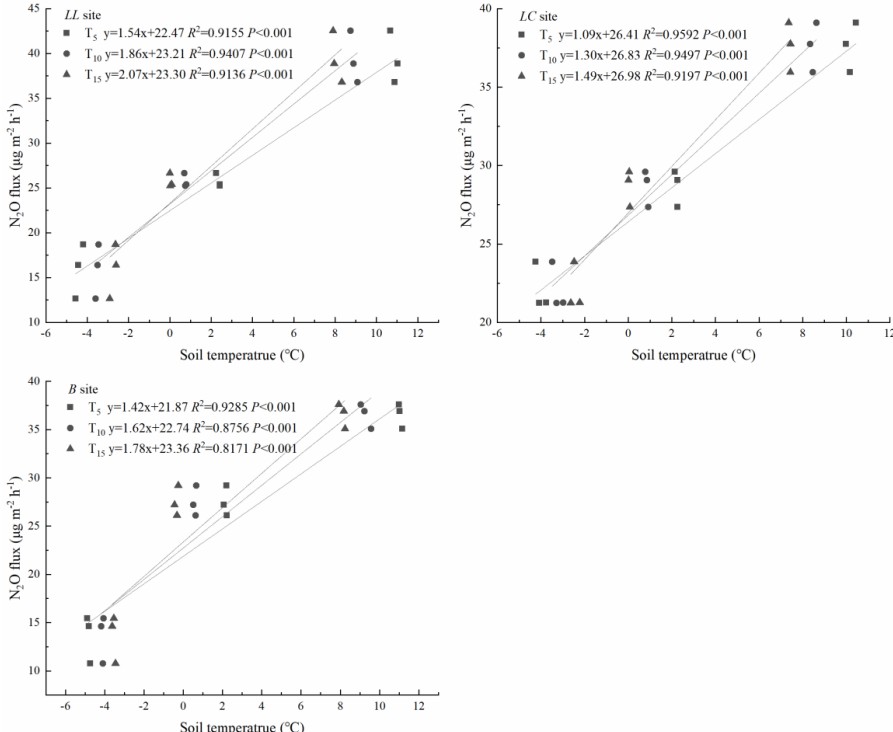


**Figure 4.** The relationship between mean $N_2O$ emission from different specified

periods and soil temperature in the permafrost region of the Daxing'an Mountains,
Northeast China.

**4.2 Contribution of nongrowing season to annual $N_2O$ budget**
The cumulative $N_2O$ emissions from the permafrost region was mainly evaluated
during the growing season (Repo et al., 2009;Takakai et al., 2008;Gao et al., 2019b).



The $N_2O$ emissions from the permafrost region were as high as 14 kg ha$^{-1}$ and
released approximately 0.1 Tg yr$^{-1}$ $N_2O$ emissions to the atmosphere in the bare peat
region of the Arctic, accounting for up to 0.6% of the global annual $N_2O$ emissions
(Repo et al., 2009). A few studies have indicated that the nongrowing season
contributed greatly to the annual $N_2O$ emissions from the frigid terrestrial ecosystems
(Li et al., 2012;Zhang et al., 2018;Fu et al., 2018). However, the contribution of the
nongrowing season to the annual $N_2O$ budget is uncertain in the permafrost region.

The cumulative $N_2O$ emissions during the nongrowing season ranged from 0.89

to 1.44 kg ha$^{-1}$, which contributed to 41.96–53.73% of the annual budget in the
permafrost region of the Daxing'an Mountains. In frigid terrestrial ecosystems, the
$N_2O$ emissions of the nongrowing season contributed to 20–74% of the annual
emissions; therefore, our results were within the range of previous studies (Li et al.,
2012;Zhang et al., 2018;Fu et al., 2018;Marushchak et al., 2011). During the spring
thaw period, the soil temperature and soil moisture dramatically changed, which
significantly affected the release of $N_2O$ emissions. A pulse or burst of $N_2O$ emissions
have been observed in agriculture (Flesch et al., 2018), grassland (Virkajärvi et al.,
2010), forest (Wu et al., 2010), marsh (Song et al., 2008), and peat ecosystems (Flesch
et al., 2018). The pulse $N_2O$ emissions during the spring thaw period had enormous
influence on the contribution of the nongrowing season to the annual budget in the
frigid terrestrial ecosystems (Li et al., 2012;Fu et al., 2018). When the pulse of $N_2O$
emissions occurred during the spring thaw period, emissions from the non-growing





seasons dominated (67–74%) the annual total emissions (Fu et al., 2018). When no
pulse of $N_2O$ emissions were found during the spring thaw period, the contribution of
the spring thaw period to the total annual $N_2O$ budget was very small and accounted
for only 6.6% of the annual emissions (Li et al., 2012). In the present study, there was
no significant large burst of $N_2O$ emissions during the spring thaw period. The
cumulative $N_2O$ emissions during the spring thaw period ranged from 0.35 to 0.66 kg
$ha^{-1}$ and contributed 15.61 to 33.00% of the annual budget in the permafrost region of
the Daxing'an Mountains, and these ranges were generally lower than the emissions
during the winter and growing season.

The mean $N_2O$ emissions were the lowest in the winter and highest in the

growing season, which were not the same as the cumulative $N_2O$ emissions in the
three swamp forests. The permafrost region of the Daxing'an Mountains was located
at a high latitude, which had a longer nongrowing season than growing season. In the
permafrost region of the subarctic, the nongrowing season lasts for more than 9
mouths (283 days) (Marushchak et al., 2011). The length of winter was more than
twice of spring thaw spring in the Daxing'an mountains. Although the mean $N_2O$
emissions in winter were lower than during spring thaw period, the cumulative $N_2O$
emissions in winter were higher than the emissions from the spring thaw period. The
cumulative $N_2O$ emissions during winter were as important as the spring thaw period.
Only one study reported the cumulative $N_2O$ emissions during nongrowing seasons in
the permafrost region. Marushchak et al. (2011) found that the $N_2O$ emissions from





the nongrowing season contributed 20–69% to the annual emissions in the bare peat
zone of the permafrost region. Our results confirmed that half of the $N_2O$ emissions
were released during the nongrowing season, indicating that the $N_2O$ emissions during
the nongrowing season cannot be ignored in the permafrost regions. In the future, the
$N_2O$ emissions of the nongrowing season should be emphasized in the permafrost
region, especially in the context of global climate change.

**4.3 Drivers of $N_2O$ emissions on different temporal scales**
Most previous studies on the permafrost region have focused exclusively on the
growing season (Gao et al., 2019b;Repo et al., 2009;Cui et al., 2018). The $N_2O$
emissions during the growing season were mainly influencing by air temperature, soil
temperature, water table level, soil moisture, precipitation, pH, $NH_4^+$-N, $NO_3^-$-N,
TOC, gross N mineralization, N content, and C/N ratio in the permafrost region (Ma
et al., 2007;Gil et al., 2017;Marushchak et al., 2011;Chen et al., 2017;Cui et al.,
2018;Paré and Bedard-Haughn, 2012). However, the drivers of nongrowing season
$N_2O$ emission remain unknown in the permafrost region.

$N_2O$ production processes are very complex and can be produced by nitrification,
nitrifier denitrification, and denitrification in the permafrost region (Ma et al.,
2007;Gil et al., 2017;Siljanen et al., 2019). Soil water content controls the redox
conditions in the soil, which determines the pathway of $N_2O$ emissions. The soil water
content showed significant temporal variation in the permafrost region of the



516 Daxing'an Mountains. Thus, the pathway of N$_2$O emissions may be different in the

517 nongrowing season and growing season. During the growing season, the water table

518 level ranged from −31.6 to 10.27 cm in the three swamp forests. N$_2$O emission may

519 have come from simultaneous nitrification and denitrification in the *LL* site and

520 denitrification in the *LC* and *BC* sites (Gao et al., 2019b). In the 2016 growing season,

521 the N$_2$O emissions mainly controlled multiple environmental factors (Gao et al.,

522 2019b). Our results show that the N$_2$O emissions were driven by soil temperature and

523 water table level and their interactions, explaining 12.36–26.35% of temporal

524 variation of N$_2$O emissions during the two growing seasons. As the permafrost

525 ecosystem is strongly temperature limited, the N$_2$O emissions were mainly

526 significantly positively correlated with soil temperature in the permafrost region

527 (Marushchak et al., 2011;Cui et al., 2018;Chen et al., 2017). Soil temperature could

528 increase nitrification and denitrification, thus promoting the release of N$_2$O flux. Wu

529 et al. (2019) found that the N$_2$O emissions were significantly negatively correlated

530 with soil moisture in the three forests of the Daxing'an Mountains. Our results

531 confirm that the decrease of the water table level was beneficial to the release of N$_2$O

532 emissions. Soil temperature and water table level were key factors controlling the

533 emission of N$_2$O during the growing season. In contrast, the nongrowing season N$_2$O

534 emissions were mainly controlled by soil temperature in the permafrost region of the

535 Daxing'an Mountains.


537 During the nongrowing season, the soil moisture was consistently exceeded 60%





in the three swamp forests. The N$_2$O emissions were major produced by
denitrification in the permafrost region. The nongrowing season N$_2$O emissions were
positively correlated with soil temperature in the three swamp forests. The denitrifier
genes were stable during the winter and increased during the spring thaw period (Tatti
et al., 2014). Wertz et al. (2013) found that the community structures of denitrifiers
were different between below-zero and above-zero temperatures. During the
nongrowing season, the soil temperature affected the abundance and community
structures of denitrifiers and thus the release of N$_2$O emissions in the permafrost
region of the Daxing'an Mountains. In the field, environmental factors are always
changing; therefore, any factor can be a limiting factor for a long period. The N$_2$O
emissions from the two-year study period were controlled by soil temperature in the
permafrost region of the Daxing'an Mountains. Soil temperature was the major
limiting factor related to annual N$_2$O emissions in the permafrost region. Except for
soil temperature, the N$_2$O emissions from the permafrost region were also affected by
pH, NO$_3^-$-N, TN, and C/N ratio.

**5 Conclusions**

The N$_2$O emissions from the nongrowing season were quantified in the

permafrost region. The N$_2$O emissions ranged from $-35.75$ to $74.16$ $\mu g \cdot m^{-2} \cdot h^{-1}$
during the nongrowing season. The mean N$_2$O emissions were lowest in the winter
and highest in the growing season, and were controlled by soil temperature. The
cumulative N$_2$O emissions during the nongrowing season greatly contributed to the



annual budget, which cannot be ignored. In the different periods studied, $N_2O$
emissions had different key limiting factors in the permafrost region. The nongrowing
season and the annual $N_2O$ emissions were driven by soil temperature, whereas the
growing season $N_2O$ emissions were affected by soil temperature, water table level,
and their interaction.

*Data availability*. The data used in the present study are available in the Supplement.

*Supplement*. The supplement related to this article is available online at:
https://doi.org/

*Author contributions*. Dawen Gao and Hong Liang designed and guided the
experiment, provided supervision, and contributed to revise of the manuscript. Tijiu
Cai and Houcai Sheng provided and set up field experimental sites. Liquan Song
conducted laboratory analysis. Weifeng Gao conducted the field sampling, analyzed
the data and wrote the manuscript.

*Competing interests*. The authors declare that they have no conflict of interest.

*Acknowledgements*. We are sincerely grateful to Heilongjiang Mohe Forest Ecosystem
Research Station.





*Financial support*. This work was supported by the National Natural Science
Foundation of China (No. 31870471, 31971468).

*Review statement*. This paper was edited by     and reviewed by.

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
