# Peer review of "Contribution of the nongrowing season to annual N2O emissions from the 1 continuous permafrost region in Northeast China 2 3 Weifeng Gao1,2, Dawen Gao1,2,3\*, Liquan Song1, Houcai Sheng4,5, Tijiu Cai4,5, Hong 4 Liang1\* 5 6 7 1School"

_Biogeosciences, 2020_

## Short Comment (SC1) · 12 Jan 2021

songcc@neigae.ac.cn;sunli@neigae.ac.cn

The authors investigated N2O emissions from the permafrost region in Northeast China from June 2016 to June 2018 using the static opaque chamber method, and analyzed the contribution of the nongrowing season to annual N2O emissions. The design of the study and interpretation of the results are appropriate. Although the topic might be suitable for publication in Biogeosciences, but the paper contains many fragments that reduplicated from the published paper of the author (Atmospheric Environment, 2019, 214, 116822), and writing/clarity needs additional attention. It is not suitable for publication in Biogeosciences in the present state. I suggest the authors revised the

paper and submit to another journal.

Detail comments:

1. Please add an aerial photo showing the sampling locations. 2. Line 24, 25, and line 51, "permafrost are thawing" not "permafrost regions are thawing". 3. Page 6 line 133, investigations? 4. Page 6 line 128, the soil type should be given according to the USDA classification system, 5. Page 8, table 1, the unit of data is missing. The meaning of small letter in the table should be given in footnote. 6. The title of the paper is Contribution of the nongrowing season to annual N2O emissions, Page 8 line 181,Gas samples were taken monthly during the winter from October to December, the time interval is too long 7. Page 11 line 212, mol/L should be mol L-1. 8. Page 13 line 265, A and B in the figure title should be a and b. 9. Page 29 line 465, release of N2O emissions. 10. Page 29 line 469, nongrowing season and line 271 non-growing seasons. 11. Page 30 line 487, mouths? 12. Page 30 line 488, spring thaw spring?

---

## Author Comment (AC1) · 16 Jan 2021

Dear Prof. Song: Thank you for your letter and the constructive comments on our manuscript entitled "Contribution of the nongrowing season to annual N2O emissions from the continuous permafrost region in Northeast China" (No.: bg-2020-305). Those comments are very helpful for revising and improving our paper. We have revised the manuscript carefully according to the comments, which we hope meet with approval. All the responses to your comments are as following:

Major comments Thank you for your approval of the design of the study and inter-pretation of the results. While, this manuscript was different with the previous paper

published in Atmospheric Environment (2019, 214, 116822). The previous published paper (Atmospheric Environment, 2019) was mainly focus on the response of N2O emissions to spring thaw period (March 17 to May 23, 2017) in the permafrost region. The authors investigated N2O emissions from a permafrost region in response to spring thaw period; explored the influence of swamp forest types on soil N2O emissions during spring thaw period; quantified the environmental drivers of N2O emissions during spring thaw period; and compared N2O emissions with those from growing season in the permafrost region and those from other ecosystems during spring thaw period at global scale. The scientists had been studied N2O emission during the growing season and spring thaw period in the permafrost region of Daxing'an mountains. However, little is known about the characteristics of nongrowing season N2O emissions from the permafrost region of Daxing'an mountains, as well as global scale of permafrost region. In this study, we measured two full-year around (June 2016 to June 2018) N2O emissions from the permafrost regions of Daxing'an mountains. We are mainly focus on the seasonal dynamics of nongrowing season N2O emissions and its contribution to the annual budget. We found that the nongrowing season N2O emissions were ranged from −35.75 to 74.16 $\mu$gÂům−2Âůh−1 in the permafrost region of Daxing'an mountains, which were mainly lower than that from growing season. The mean soil temperature controlled the mean N2O emissions from growing season, spring thaw period and nongrowing season. The cumulative N2O emissions from nongrowing season contributed to 41.96–53.73% of the annual budget, accounting for almost half of the annual emissions in the permafrost region, which was an important component of annual emissions and cannot be ignored in the permafrost region. Our results proved the observation of the nongrowing season N2O emissions from permafrost regions is of great significance for the accurate assessment of regional N2O emission in the permafrost regions.

Detail comments 1. Please add an aerial photo showing the sampling locations. The authors' answer: Thank you for your meaningful suggestion. We have added the vegetation and sampling location photo.

2. Line 24, 25, and line 51, "permafrost are thawing" not "permafrost regions are thawing". The authors' answer: Thank you for your comment. We have replaced the "permafrost regions are thawing" with "permafrost are thawing".

3. Page 6 line 133, investigations? The authors' answer: We have revised it. We used the investigate instead of investigations.

4. Page 6 line 128, the soil type should be given according to the USDA classification system, The authors' answer: Thank you for your comment. The soil type were Gelisols according to the USDA classification system. We have revised it.

5. Page 8, table 1, the unit of data is missing. The meaning of small letter in the table should be given in footnote. The authors' answer: Thank you for your meaningful suggestion. We have added the unit of the soil physical and chemical factors. In the footnote, we added the meaning of small letter in the table.

6. The title of the paper is Contribution of the nongrowing season to annual N2O emissions, Page 8 line 181, Gas samples were taken monthly during the winter from October to December, the time interval is too long The authors' answer: Thank you for your meaningful suggestion. It's true that the time interval in winter was a little long. In 2019, we measured the N2O emissions during the autumn freeze-thaw period (September 27 to November 11) (Zeng, 2020). The autumn freeze-thaw period was an important part of the winter. During the autumn freeze-thaw period, environmental factors were changed dramatically, which would release a large amount of N2O emissions same as the spring freeze-thaw period. However, the results show that the mean N2O emissions were $2.32 \pm 4.92$, $3.37 \pm 4.10$ and $1.06 \pm 2.64$ $\mu$g m$-2$ h$-1$ from LL, LC and BC sites, respectively (Zeng, 2020). The spatial variation of N2O emissions were very small. T We didn't measure the N2O emissions from middle of November to the end of December. We speculate that the change of N2O emissions would very small due to the soil was freezing. In the future, we would add the sampling frequency of the winter. Reference Zeng QB. Analysis of greenhouse gas flux and global warming

potential in different permafrost zones of Daxing'an mountains. Harbin: Harbin Institute of Technology, 2020. (In Chinese)

7. Page 11 line 212, mol/L should be mol L-1. The authors' answer: We have replaced mol/L with mol L−1.

8. Page 13 line 265, A and B in the figure title should be a and b. The authors' answer: Thank you for your comment. We have revised it.

9. Page 29 line 465, release of N2O emissions. The authors' answer: We have deleted the "released of".

10. Page 29 line 469, nongrowing season and line 271 non-growing seasons. The authors' answer: Thank you for your suggestion. We have revised it.

11. Page 30 line 487, mouths? The authors' answer: That's my spelling mistake. We have revised it.

12. Page 30 line 488, spring thaw spring? The authors' answer: Thank you for your comment. We have replaced spring thaw spring with spring thaw period.

Thank you for your comments; we are happy to make additional revisions if needed.

Sincerely, The authors,

Dr. Da-Wen Gao, School of Environment and Energy Engineering, Beijing University of Civil Engineering and Architecture, Beijing 100044, China E-mail address: gaodw@hit.edu.cn

———————————————————

(a) *LL* swamp forest

(b) *LC* swamp forest

(c) *BC* swamp forest

(d) Location of the study site

**Fig. 1.** The vegetation (a, b and c) and location (d) of the study site in the Daxing'an Mountains, northeast China.

---

## Referee Comment (RC1) · Anonymous Referee #1 · 21 Apr 2021

The manuscript (MS) investigated N2O emission from swamp forest soils in permafrost region during the nongrowing season, and evaluated its contribution to annual N2O emission. Since permafrost regions have long winter periods, the importance of assessing N2O emissions during winter is understandable. The theme of this MS seems to be within the scope of BG. However, there are some serious concerns on this MS, especially for measurements, presentation of data, and discussion. So, I think that fundamental revisions are necessary for the publication of this MS. Please consider the comments below for your revision of the manuscript.

[General comments] 1. This MS evaluates the contribution of nongrowing season to

annual N2O emissions, but during the winter period, N2O emission was not measured for almost the entire period from December to February. Although the authors state that winter period is longer than soil thawing period and therefore has more cumulative N2O emission (L482-490), there are no measurements that adequately cover this period. By linear interpolation, it is estimated that N2O emission continues to occur during this unmeasured period (Figure 1). However, could significant N2O emission occur during soil freezing? The authors should clearly explain the question of the legitimacy of the integrated release estimate caused by the lack of frequency of measurements.

2. Although the MS focuses on N2O emissions during the nongrowing season, there are many descriptions that focus on the growing season (e.g., L295-298, 511-532); the discussion should be substantially reconstructed to focus on the description of nongrowing season.

3. "N2O emission is low in winter because the temperature is low". To state this, there is no need for redundant discussion as in this MS. Figures 3 and 4 are a rehash of the data presented in the previous section, but there is no significance in averaging the N2O emissions for each period and verifying the correlation with temperature again. Throughout the discussion, there are many overlapping statements. Environmental factors other than temperature are almost completely absent from the discussion. Although measurements were taken at three sites, there is no comparison between the sites. In light of the above, the discussion should be thoroughly restructured.

[Specific comments] L120-129: It is described as a "permafrost region", but there is almost no information about permafrost (e.g., thickness of permafrost layer, active layer depth, soil thawing period, etc.) L170-171: Did you place the collar in a different location for each measurement? L201-205: Were soil samples taken for each gas measurement? Figure 1: What do the error bars indicate? L310-334: Soil C/N, TOC, and TN have been shown to be controlling factors for the temporal variations of N2O emissions, but do these values change over time like N2O emissions? In this analysis, is there any spatial variation between iterations mixed in with the temporal variation? To

verify the temporal variations, we should average the replications and then correlate them with environmental factors. In addition, the seasonal changes in environmental factors are not shown, so it is difficult to judge the correlation. L403-412: I think it should be shown in the results.

Sincerely yours,

---

## Author Comment (AC2) · 7 May 2021

Dear Referee:

Thank you for your letter and the constructive comments on our manuscript entitled "Contribution of the nongrowing season to annual N2O emissions from the continuous permafrost region in Northeast China" (No.: bg-2020-305). The comments are very helpful for improving our paper, as well as the important guiding significance to our research. We have revised the manuscript carefully according to the comments, which we hope meet with approval. Major modifications in the revised manuscript were noted

as yellow. All the responses to your comments are as following:

General comments

1. This MS evaluates the contribution of nongrowing season to annual N2O emissions, but during the winter period, N2O emission was not measured for almost the entire period from December to February. Although the authors state that winter period is longer than soil thawing period and therefore has more cumulative N2O emission (L482-490), there are no measurements that adequately cover this period. By linear interpolation, it is estimated that N2O emission continues to occur during this unmeasured period (Figure 1). However, could significant N2O emission occur during soil freezing? The authors should clearly explain the question of the legitimacy of the integrated release estimate caused by the lack of frequency of measurements.

The authors' answer:

Thank you for your meaningful suggestion. Previous studies reported that freezing-thawing cycles (include spring thaw period and autumn freezing period) had a significantly influence on the N2O emissions. We are also concerned about this issue. During this study period, we reported the response of N2O emissions to spring thaw period in the permafrost region. We found that there were no significantly burst or pulse of N2O emissions (range from $-35.75$ to $74.16$ $\mu$g m$-2$ h$-1$) as observed in other ecosystems (Gao et al., 2018). From 27 September to 11 November 2019, we measured the N2O emission every week during the autumn freezing period. The results showed that the N2O emissions were ranged from $-4.21$ to $12.86$ $\mu$g m$-2$ h$-1$ during the autumn freezing season (Unpublished). The N2O emissions were not significantly released during the soil freezing. Indeed, the sampling of N2O emissions during the winter were lack. According your meaningful suggestion, we clearly explained the question of the legitimacy of the integrated release estimate caused by the lack of frequency of measurements. In brief, the temporal variation of N2O emission during the nongrowing season were relatively stable in the permafrost-affect soils (Pei et al., 2004; Du et al.,

2016; Cao et al., 2018; Du et al., 2008; Kato et al., 2013). Significantly N2O emissions were only observed in October (autumn freezing period) in the subarctic (Marushchak et al., 2011). Thus, we estimated that the N2O emissions during the nongrowing season in the Daxing'an Mountains were relatively stable like previous studies (Pei et al., 2004;Du et al., 2016;Cao et al., 2018;Du et al., 2008;Kato et al., 2013). Please see L455-471.

Reference:

Cao, Y. F., Ke, X., Guo, X. W., Cao, G. M., and Du, Y. O.: Nitrous oxide emission rates over 10 years in an alpine meadow on the Tibetan Plateau, Pol J Environ Stud, 27, 1353-1358, https://doi.org/10.15244/pjoes/76795, 2018. Du, Y., Cui, Y., Xu, X., Liang, D., Long, R., and Cao, G.: Nitrous oxide emissions from two alpine meadows in the Qinghai-Tibetan Plateau, Plant Soil, 311, 245-254, https://doi.org/10.1007/s11104-008-9727-9, 2008. Du, Y. G., Guo, X. W., Cao, G. M., Wang, B., Pan, G. Y., and Liu, D. L.: Simulation and prediction of nitrous oxide emission by the water and nitrogen management model on the Tibetan plateau, Biochem Syst Ecol, 65, 49-56, https://doi.org/10.1016/j.bse.2016.02.002, 2016. Gao, W. F., Yao, Y. L., Gao, D. W., Wang, H., Song, L. Q., Sheng, H. C., Cai, T. J., and Liang, H.: Responses of N2O emissions to spring thaw period in a typical continuous permafrost region of the Daxing'an Mountains, northeast China, Atmos Environ, 214, 116822, https://doi.org/10.1016/j.atmosenv.2019.116822, 2019. Kato, T., Toyoda, S., Yoshida, N., Tang, Y. H., and Wada, E.: Isotopomer and isotopologue signatures of N2O produced in alpine ecosystems on the Qinghai-Tibetan Plateau, Rapid Commun Mass Sp, 27, 1517-1526, https://doi.org/10.1002/rcm.6595, 2013. Marushchak, M. E., Pitkamaki, A., Koponen, H., Biasi, C., Seppala, M., and Martikainen, P. J.: Hot spots for nitrous oxide emissions found in different types of permafrost peatlands, Global Change Biol, 17, 2601-2614, https://doi.org/10.1111/j.1365-2486.2011.02442.x, 2011. Pei, Z. Y., Ouyang, H., Zhou, C. P., and Xu, X. L.: N2O exchange within a soil and atmosphere profile in alpine grasslands on the Qinghai-Xizang Plateau, Acta Bot Sin, 46, 20-28,

https://doi.org/10.1046/j.1365-3180.2003.00373.x, 2004.

2. Although the MS focuses on N2O emissions during the nongrowing season, there are many descriptions that focus on the growing season (e.g., L295-298, 511-532); the discussion should be substantially reconstructed to focus on the description of nongrowing season.

The authors' answer:

Thank you for your meaningful suggestion. In the manuscript, we reconstructed the discussion. The description of discussion during the growing season were reduced. We fucus on the description of nongrowing season N2O emissions and highlighting the importance of N2O emissions during the nongrowing season.

3. "N2O emission is low in winter because the temperature is low". To state this, there is no need for redundant discussion as in this MS. Figures 3 and 4 are a rehash of the data presented in the previous section, but there is no significance in averaging the N2O emissions for each period and verifying the correlation with temperature again. Throughout the discussion, there are many overlapping statements. Environmental factors other than temperature are almost completely absent from the discussion. Although measurements were taken at three sites, there is no comparison between the sites. In light of the above, the discussion should be thoroughly restructured.

The authors' answer:

Thank you for your meaningful suggestion. We deleted the discussion of the difference of N2O emissions among different periods. We revised and deleted the overlapping statements. According you suggestion, we discussed the difference of nongrowing season N2O emissions among the three swamp forest types. Except for soil temperature, we discussed the effect of other environment factors on the N2O emissions. Please see Discussion 4.1 and 4.3.

Specific comments

1. L120-129: It is described as a "permafrost region", but there is almost no information about permafrost (e.g., thickness of permafrost layer, active layer depth, soil thawing period, etc.)

The authors' answer: Thank you for your meaningful suggestion. We have added the information about the permafrost. Please see L123-126.

2. L170-171: Did you place the collar in a different location for each measurement?

The authors' answer: No, the collars were permanently inserted into the soil during the whole study period. On each measurement, the chamber was placed on the collar and filled with water to collect $N_2O$ emitted from the soil.

3. L201-205: Were soil samples taken for each gas measurement?

The authors' answer: Yes. During each gas measurement, the soil sample were taken close to each collar except for the spring thaw period in 2017. The $N_2O$ emissions were measured every three to ten days during the spring thaw period, but the soil samples were taken every ten days. Gas samples were collected 45 times and soil samples were collected 38 times. The temporal variation of environment factors was shown in Fig. 2.

4. Figure 1: What do the error bars indicate?

The authors' answer: The error bars were standard deviation (SD). We have added it in the Figure and tables.

5. L310-334: Soil C/N, TOC, and TN have been shown to be controlling factors for the temporal variations of $N_2O$ emissions, but do these values change over time like $N_2O$ emissions?

The authors' answer: During the entire measurement, there had temporal variation on the soil C/N, TOC, and TN. We added the description and the figure of seasonal changes in environmental factors. Please see L227-259.

6. In this analysis, is there any spatial variation between iterations mixed in with the temporal variation? To verify the temporal variations, we should average the replications and then correlate them with environmental factors.

The authors' answer: Thank you for your meaningful suggestion. We used Pearson's correlation analysis, linear correlation analysis, and multivariate regression analysis to analyze the relationship between N2O emissions and corresponding environmental factors from each collar. According to your suggestion, we averaged the replications and then correlate them with environmental factors. After the average, the amount of data were small, and partial results became no significantly correlations. The analysis used in the manuscript would be more significantly.

7. In addition, the seasonal changes in environmental factors are not shown, so it is difficult to judge the correlation. L403-412: I think it should be shown in the results.

The authors' answer: Thank you for your meaningful suggestion. In the result section, we added the description and the Figure of seasonal changes in environmental factors. Meanwhile, we added the difference of soil environment factors among the three type of swamp forests (Table 1). Please see L227-275. According to your suggestion, we put the L403-412 to the results section.
* * *
Dr. Da-Wen Gao, School of Environment and Energy Engineering, Beijing University of Civil Engineering and Architecture, Beijing 100044, China E-mail address: gaodw@hit.edu.cn

---

## Referee Comment (RC2) · Lutz Merbold (Referee) · 18 May 2021

The authors present a study that investigates N2O emissions form three swamp forests in permafrost region of NorthEast China. Specifically the authors aim at addressing the contribution of non-growing season N2O emissions to the annual budget - clearly a challenge to derive reliable data with sufficient temporal resolution in permafrost regions.

While the aim of the study become clear, the authors tend to "oversell" their results in various places in the manuscript. The fact that they focus on the swamp forests only
comes out at a late stage while the title suggests something very different. While the could be of potential interest to the readers of BG, the current version of the manuscript can not accepted for publication for various reasons.

Overall, the results cover both, growing and non-growing season data. The non-growing season data has considerably less temporal coverage - while it also remains unclear how long the actual growing season lasts - yet no uncertainty estimates given the accumulated numbers. At the same time, the analysis of driver variables is superficial and needs considerable work. The discussion is a loose list, sometimes a chaotic list of studies and what these found, thus it is extremely difficult for the reader to know, whether the numbers and facts presented are part of this study or another study. The actual discussion of the results however is lacking.

Some more minor comments, which need to be addressed nevertheless: No hypothesis given, The conclusion is a repetition of the results, Gapfilling procedures to derive annual budgets are not explained in detail, and many more which can be found in the commented PDF file.

While this may not be the answer the authors would like to receive, I would like to encourage them to take the time to work on the manuscript following the suggestions provided and possibly submit to BG again.

with kind regards

Lutz Merbold

Please also note the supplement to this comment:
https://bg.copernicus.org/preprints/bg-2020-305/bg-2020-305-RC2-supplement.pdf

**Supplement:**

22

[revised manuscript text omitted]
                                              | $-13.19 \pm 9.73b$         | $-4.51 \pm .51a$    | $0.26 \pm 5.67a$    |
| SM 0-10                               | $117.30 \pm 14.92c$        | $174.20 \pm 14.58a$ | $162.31 \pm 16.14b$ |
| SM 10-20                              | 49.54 ± 8.28b              | $115.86 \pm 10.98a$ | 115.91 ± 9.13a      |
| pH 0-10                               | $4.77 \pm 0.16c$           | $4.99 \pm 0.08a$    | $4.89 \pm 0.11b$    |
| pH 10-20                              | $4.93 \pm 0.18c$           | $5.09 \pm 0.07a$    | $4.99 \pm 0.11b$    |
| NH4 + -N0-10                          | $5.49 \pm 2.15a$           | $5.98 \pm 3.03a$    | $4.92 \pm 2.65a$    |
| NH4 + -N10-20                         | $3.05 \pm 1.57a$           | $3.87 \pm 1.94a$    | $3.43 \pm 1.88a$    |
| NO 3 - -N 0-10  | $1.71 \pm 0.73a$           | $1.81 \pm 1.02a$    | $1.58 \pm 0.63a$    |
| NO 3 - -N 10-20 | $1.29 \pm 0.57 ab$         | $1.44 \pm 1.02a$    | $1.02 \pm 0.42b$    |
| TOC 0-10                              | <mark>39.95</mark> ± 6.91a | $42.01 \pm 4.43a$   | $35.57 \pm 5.22 b$  |
| TOC 10-20                             | $15.62\pm3.95b$            | $18.25\pm2.71a$     | 16.62 ± 2.1ab       |
| $TN_{0-10}$                                      | $2.19\pm0.37\text{b}$      | 3.78 ± 0.51a        | $1.97\pm0.69b$      |
| TN 10-20                              | $0.83 \pm 0.15 b$          | $1.03 \pm 0.21a$    | $0.91 \pm 0.13 b$   |
| C/N 0-10                              | $17.94 \pm 4.17a$          | $11.27 \pm 144b$    | $17.08\pm3.55a$     |
| C/N 10-20                             | $19.26\pm5.65a$            | $18.29 \pm 4.24a$   | $18.57\pm3.82a$     |

158

WTL, water table level; SM0-10, soil moisture at 0-10 cm; SM10-20, soil moisture 159 at 10-20 cm; pH0-10, pH at 0-10 cm; pH10-20, pH at 10-20 cm; NH4+-N0-10, ammonium nitrogen at 0-10 cm; NH4+-N10-20, ammonium 
[revised manuscript text omitted]

---

## Author Comment (AC3) · 6 Jun 2021

Dear Referee:

After receive your comments, we revised the manuscript as soon as possible. And submitted the reply letter on the official website of Biogeosciences. However, we revised the manuscript again based on the Dr. Lutz's comments. Therefore, the line numbering in the reply letter were different with final version of the manuscript. We have corrected the line numbering. The specific amendments are as follows:

General comments
1. This MS evaluates the contribution of nongrowing season to annual N2O emissions, but during the winter period, N2O emission was not measured for almost the entire period from December to February. Although the authors state that winter period is longer than soil thawing period and therefore has more cumulative N2O emission (L482-490), there are no measurements that adequately cover this period. By linear interpolation, it is estimated that N2O emission continues to occur during this unmeasured period (Figure 1). However, could significant N2O emission occur during soil freezing? The authors should clearly explain the question of the legitimacy of the integrated release estimate caused by the lack of frequency of measurements.

The authors' answer:

The line numbering should be 455-471.

Specific comments

1. L120-129: It is described as a "permafrost region", but there is almost no information about permafrost (e.g., thickness of permafrost layer, active layer depth, soil thawing period, etc.)

The authors' answer:

The line numbering should be 132-135.

5. L310-334: Soil C/N, TOC, and TN have been shown to be controlling factors for the temporal variations of N2O emissions, but do these values change over time like N2O emissions?

The authors' answer:

The line numbering should be 258-271.

7. In addition, the seasonal changes in environmental factors are not shown, so it is difficult to judge the correlation. L403-412: I think it should be shown in the results.

The authors' answer:

The line numbering should be 240-285.

Thank you for your comments; we are happy to make additional revisions if needed.

Sincerely,

The authors,

Dr. Da-Wen Gao,

School of Environment and Energy Engineering, Beijing University of Civil Engineering and Architecture, Beijing 100044, China

E-mail address: gaodw@hit.edu.cn

---

## Author Comment (AC4) · 6 Jun 2021

Dear Lutz Merbold:

Thank you for your letter and the constructive comments on our manuscript entitled "Contribution of the nongrowing season to annual N2O emissions from the continuous permafrost region in Northeast China" (No.: bg-2020-305). Those comments are very helpful for revising and improving our paper, as well as the important guiding significance to our research. We have carefully checked the paper and revised it base on your comments, which we hope meet with approval. Major modifications in the revised
manuscript were noted as yellow. All the responses to your comments are as following:

General remarks and major comments:

The authors present a study that investigates N2O emissions form three swamp forests in permafrost region of NorthEast China. Specifically the authors aim at addressing the contribution of non-growing season N2O emissions to the annual budget - clearly a challenge to derive reliable data with sufficient temporal resolution in permafrost regions.

While the aim of the study become clear, the authors tend to their results in various places in the manuscript. The fact that they focus on the swamp forests only comes out at a late stage while the title suggests something very different. While the could be of potential interest to the readers of BG, the current version of the manuscript can not accepted for publication for various reasons.

Overall, the results cover both, growing and non-growing season data. The nongrowing season data has considerably less temporal coverage - while it also remains unclear how long the actual growing season lasts - yet no uncertainty estimates given the accumulated numbers. At the same time, the analysis of driver variables is superficial and needs considerable work. The discussion is a loose list, sometimes a chaotic list of studies and what these found, thus it is extremely difficult for the reader to know, whether the numbers and facts presented are part of this study or another study. The actual discussion of the results however is lacking.

Some more minor comments, which need to be addressed nevertheless: No hypothesis given, The conclusion is a repetition of the results, Gapfilling procedures to derive annual budgets are not explained in detail, and many more which can be found in the commented PDF file.

Thank you for your meaningful suggestion. According to your suggestion, we have carefully revised the manuscript.

First, indeed, obtaining reliable data with sufficient temporal resolution in permafrost regions is clearly a challenge. Voigt et al., (2020) summarized the studies on the N2O emissions from permafrost-affected soils, which found that the measurements of N2O emissions in permafrost regions were sparse and lacking during the nongrowing season, making the magnitude of N2O fluxes across the vast permafrost regions uncertain. Lack of wintertime measurements from permafrost regions adds to the uncertainty of the estimate contribution of nongrowing season emissions to annual budget (Voigt et al., 2020). Thus, we hope to fill in the data on the temporal variation of N2O emissions during the nongrowing season, the contribution of N2O emissions from the nongrowing season to annual budget, and the key regulatory factors on N2O emissions during the nongrowing season in this study. Indeed, sampling frequency of N2O emissions was lack during the winter in our study. Only one study reported the contribution of nongrowing season N2O emission to annual budget in the permafrost region, which was also lack of sampling frequency during the nongrowing season (Marushchak et al., 2011). According to previous studies, we clearly explained the question of the legitimacy of the integrated release estimate caused by the lack of frequency of measurements. Please see lines 469-486.

Reference

Marushchak, M. E., Pitkamaki, A., Koponen, H., Biasi, C., Seppala, M., and Martikainen, P. J.: Hot spots for nitrous oxide emissions found in different types of permafrost peatlands, Global Change Biol, 17, 2601-2614, https://doi.org/10.1111/j.1365-2486.2011.02442.x, 2011.

Voigt, C., Marushchak, M. E., Abbott, B. W., Biasi, C., Elberling, B., Siciliano, S. D., Sonnentag, O., Stewart, K. J., Yang, Y., and Martikainen, P. J.: Nitrous oxide emissions from permafrost-affected soils, Nature Reviews Earth & Environment, 1, 420-434, https://doi.org/10.1038/s43017-020-0063-9, 2020.

Second, we focused on the N2O emissions from the permafrost region. Thus, three

typical swamp forests in the permafrost region of Daxing'an Mountains were selected as our research target (Gao et al., 2019a, 2019b). The typical permafrost ecosystems in the Daxing'an mountains were different with other permafrost region, such as peatland in the arctic, tundra in Ny-Ålesund, forest in Eastern Siberia, alpine meadow in Qinghai-Tibetan Plateau. So, we thought that the $N_2O$ emissions from swamp forests were representative in the permafrost regions of Daxing'an Mountains. The results were "overall" in various places in the manuscript. According to your suggestion, we have revised it.

Reference

Gao, W. F., Yao, Y. L., Gao, D. W., Wang, H., Song, L. Q., Sheng, H. C., Cai, T. J., and Liang, H.: Responses of $N_2O$ emissions to spring thaw period in a typical continuous permafrost region of the Daxing'an Mountains, northeast China, Atmos Environ, 214, 116822, https://doi.org/10.1016/j.atmosenv.2019.116822, 2019a.

Gao, W. F., Yao, Y. L., Liang, H., Song, L. Q., Sheng, H. C., Cai, T. J., and Gao, D. W.: Emissions of nitrous oxide from continuous permafrost region in the Daxing'an Mountains, Northeast China, Atmos Environ, 198, 34-45, https://doi.org/10.1016/j.atmosenv.2018.10.045, 2019b.

Third, according to your meaningful suggestion, we defined the nongrowing season (include winter and spring thaw period) and growing season in the statistical analysis section. Please see lines 219-226. The length of nongrowing season and growing season were shown in the table 2. Please see lines 321-324. We explained the legitimacy of the integrated release estimate caused by the less temporal coverage during the nongrowing season. Please see lines 469-486. The cumulative $N_2O$ emissions were linearly and sequentially accumulated from the emissions between every two adjacent intervals of the measurements with each collar. And then the standard deviation is used to express the degree of dispersion of the cumulative $N_2O$ emissions. We re-analyzed the relationship between $N_2O$ emissions and driver variables during the growing season. We used stepwise multiple linear regression replace the multivariate regression analysis during the growing season. Please see table 5. We also rewritten the relevant abstract (Please see lines 42-45), result (Please see lines 401-410), discussion (Please see lines 559-573), and conclusion section (Please see lines 598-599). According to your suggestion, we have made major revised of the discussion. We have rewritten the 4.1 section of the discussion. We also modified the logical structure of the 4.2 and 4.3 section. Meanwhile, we clearly described the numbers and facts presented are this study or cited papers.

Fourth, according to your minor comments, we have carefully revised the manuscript. The reply is as follows:

Miner comments

Abstract

1. L32, nongrowing season N2O emissions, how is this defined?

The authors' answer:

The nongrowing season N2O emissions were defined as the N2O emissions during the nongrowing season, which were the release of N2O emissions started the daily mean air temperature remained below 0 °C for at least five consecutive days and ended at soil thawing to a depth of 20 cm. We have added the definition. Please see lines 220-226.

2. L35 1.76-2.86 times, thats very accurate, does your methods allow such accurate estimate?

The authors' answer:

Thank you for your comments. We calculated the mean N2O emissions during the winter, spring thaw period, and growing season for two years. And then we compared the mean N2O emissions among winter, spring thaw period, and growing season. We

think the data was accurate.

3. L38-39, The nongrowing season N2O emissions ranged from 0.89 to 1.44 kg ha–1, which contributed to 41.96–53.73% of the annual budget. if your growing season fluxes are that much higher, why would the contribution of nongrowing season fluxes be so large? definition and length of both seasons is essential!

The authors' answer:

Thank you for your meaningful suggestion. We defined the nongrowing season (include winter and spring thaw period) and growing season in the statistical analysis section. Please see lines 219-226. The nongrowing season lasted for about 230 days, which were 1.8-fold than that of the growing season (130 days, Table 2). The cumulative N2O emissions during the short spring thaw period were contributed to 15.63–33.00% of the annual emissions. Although the rates of N2O during the growing season were higher than that of nongrowing season, the longer nongrowing season period made the cumulative nongrowing season N2O emissions close to the emissions during the growing season.

4. L40, different among during, wording.

The authors' answer:

Thank you for your comments. We have revised it.

5. L41-44, The N2O emissions from total two-year observation period and nongrowing season were mainly affected by soil temperature, while the N2O emissions from growing season were controlled by soil temperature, water table level, and their interactions. can you be more precisely? how much of the variability of N2O fluxes were expl. by Tsoil?

The authors' answer:

Thank you for your meaningful suggestion. The N2O emissions from nongrowing season and total two-year observation period were mainly affected by soil temperature, which could explain 3.01-9.54% and 6.07-14.48% temporal variation of N2O emissions, respectively. While the N2O emissions from growing season were controlled by soil temperature, water table level, pH, NH4+-N, NO3—N, total nitrogen, total organic carbon, and C/N ratio, which could explain 14.51–45.72% temporal variation of N2O emissions. We have revised it. Please see lines 40-45.

6. L45, is an important, are an important.

The authors' answer:

We have revised it.

Introduction

1. L64, tropical soil, soils.

The authors' answer:

We have revised it.

2. L78, 40 kg ha−1, I presume you mean N per year - please adjust.

The authors' answer:

Thank you for your suggestion. We replace the kg ha−1 with kg N ha−1.

3. L116, hypothesis are missing

The authors' answer:

We have added the hypothesis. We hypothesis that: (i) N2O emissions from permafrost region had significantly temporal variation on an annual scale, and mean N2O emissions during the nongrowing season were lower than that of growing season; (ii) the nongrowing season N2O emissions had significance contribute to annual budget, which cannot be ignored in the permafrost region; (iii) there had difference key regulatory factors for N2O emissions during the nongrowing season, growing season, and

entire period. Please see lines 109-114.

Material and Methods

1. L128, primarily brown forest soil, WRB Soil type classification.

The authors' answer:

Thank you for your suggestion. According to the SC1 comments, the USDA soil type classification was used in the manuscript. The zonal soil at the study site is Gelisols (according to USDA soil classification). Please see lines 131-132.

2. L157, units are missing.

The authors' answer:

Thank you for your meaningful comments. We had added the units. Please see table 1.

3. L171, inserted 20 cm into the soil, thats quite deep - why?

The authors' answer:

The base collar was inserted 20 cm into the soil to prevent the exchange of N2O gas inside and outside of base collar, which could better evaluate the release of N2O inside the base collar.

4. L185, gas N2O concentration, N2O concentration.

The authors' answer:

We have revised it.

5. L198-199, In the nongrowing season, soil moisture was determined by the oven-drying mothed. gravimetric water content.

The authors' answer:

Yes, gravimetric water content was determined by the oven-drying mothed. And then volumetric water content was converted by gravimetric water content and bulk density. We have revised it. Please see lines 185-186 and 192-198.

6. L233, reported by (Hou et al., 2012), formatting.

The authors' answer:

We have revised it (by Hou et al., (2012)).

Results

1. L254, temporal pattern, patterns.

The authors' answer:

We have replaced pattern with patterns.

2. L260-261, Negative emissions mainly occurred during the winter and spring thaw period. Negative values were real and not part due to leaks? What were your tests to ensure chamber closure?

The authors' answer:

The negative values were real. We checked the air tightness of each chamber before gas sampling to ensure that the chamber was closure.

3. L264, Fig 1, only site three is in the end significantly different, isnt it? there is a low coverage of the winter fluxes.

The authors' answer:

Yes. The site three in the end is significantly different. In the revised manuscript, we increased the discussion of sampling frequency of N2O emissions during the winter. Please see lines 469-486.

4. L297, overall, how confident are you in the cumulative winter emissions? much lower

sampling frequency and thus larger uncertainty - see BArton et al. Scientific reports on sampling frequency vs annual/seasonal budgets. I am missing a thorough uncertainty analysis.

The authors' answer:

Thank you for your meaningful suggestion. Indeed, the sampling frequency would influence the assess of the cumulative N2O emissions. According your suggestion, we increased the discussion of N2O emission during the winter. According to your meaningful suggestion, we clearly explained the question of the legitimacy of the integrated release estimate caused by the lack of frequency during the winter. Please see lines 469-486.

5. L307, STP: spring thaw period. what does this number mean?

The authors' answer:

The number means the contribution of spring thaw period to the annual N2O emissions. Spring thaw period have a significant effect on the N2O emissions. During spring thaw period, the N2O emissions reached 15106 $\mu$g m$-2$ h$-1$ from agricultural ecosystems (Flesch et al., 2018) and the cumulative N2O emissions reached 26 kg N ha$-1$ over a 35 d spring thaw period (Dunmola et al., 2010), which could account for more than 70% of annual N2O emissions (Fu et al., 2018; Wolf et al., 2010; Wu et al., 2010; Yang et al., 2015). Thus, the cumulative N2O emission from spring thaw period was an important part of the annual budget. However, the cumulative N2O emissions during the nongrowing season were contribution 15.63-33.00% to annual N2O emissions in the permafrost region, which was not significantly as previous studies.

Reference

Dunmola, A. S., Tenuta, M., Moulin, A. P., Yapa, P., and Lobb, D. A.: Pattern of greenhouse gas emission from a Prairie Pothole agricultural landscape in Manitoba, Canada, Can J Soil Sci, 90, 243-256, https://doi.org/10.4141/cjss08053, 2010.

Flesch, T. K., Baron, V. S., Wilson, J. D., Basarab, J. A., Desjardins, R. L., Worth, D., and Lemke, R. L.: Micrometeorological measurements reveal large nitrous oxide losses during spring thaw in Alberta, Atmosphere, 9, 128, https://doi.org/10.3390/atmos9040128, 2018.

Fu, Y. F., Liu, C. Y., Lin, F., Hu, X. X., Zheng, X. H., Zhang, W., and Ca, G. M.: Quantification of year-round methane and nitrous oxide fluxes in a typical alpine shrub meadow on the Qinghai-Tibetan Plateau, Agr Ecosyst Environ, 255, 27-36, 10.1016/j.agee.2017.12.003, 2018.

Wolf, B., Zheng, X. H., Brueggemann, N., Chen, W. W., Dannenmann, M., Han, X. G., Sutton, M. A., Wu, H. H., Yao, Z. S., and Butterbach-Bahl, K.: Grazing-induced reduction of natural nitrous oxide release from continental steppe, Nature, 464, 881-884, 10.1038/nature08931, 2010.

Wu, X., Brueggemann, N., Gasche, R., Shen, Z. Y., Wolf, B., and Butterbach-Bahl, K.: Environmental controls over soil-atmosphere exchange of N2O, NO, and CO2 in a temperate Norway spruce forest, Global Biogeochem Cy, 24, GB2012, https://doi.org/10.1029/2009GB003616, 2010.

Yang, X. M., Chen, H. Q., Gong, Y. S., Zheng, X. H., Fan, M. S., and Kuzyakov, Y.: Nitrous oxide emissions from an agro-pastoral ecotone of northern China depending on land uses, Agr Ecosyst Environ, 213, 241-251, 10.1016/j.agee.2015.08.011, 2015.

6. L315-317, positively correlated with air temperature ($P<0.05$) and C/N0–10 ($P<0.05$) and negatively correlated with pH0–10 ($P<0.01$), pH10–20 ($P<0.05$), NO3−-N0–10 ($P<0.05$), NO3−-N10–20 ($P<0.05$), and TN0–10 ($P<0.05$). how did you test the effects? individually or combined?

The authors' answer:

The Pearson's correlation analysis was used to test the correlations between the N2O emissions and environmental factors. The data were individually, not combined.

7. L351, Fig. 2, still bery low explanatory power.... what about the very low values, which time period is this?

The authors' answer:

Yes, the adjust R2 were low. The soil temperature could explain 3.01-3.27% temporal variation of N2O emissions from LL swamp forest during the nongrowing season.

8. L365, Table 5, wouldn't a season specific analysis be useful? lots of parameters but low explanatory power, which driver is missing...?

The authors' answer:

Thank you for your meaningful suggestion. According to your suggestion, we used the stepwise multiple linear regression analyses to quantify the relationship between N2O emissions and environmental factors during the growing season. The explanatory power of the stepwise multiple linear regression was higher than that of multivariate regression. Please see table 5. We also revised the relevant abstract (Please see lines 42-45), result (Please see lines 401-410), discussion (Please see lines 559-573), and conclusion section (Please see lines 598-599).

Discussion

1. L370, 4.1 Soil temperature controls the mean N2O emissions from the different periods, wouldn't a season specific analysis be useful?

The authors' answer:

Thank you for your meaningful suggestion. The anonymous referee 1 agree with your comments that the soil temperature controls the mean N2O emissions from the different periods is no need for discussion. We have restructured the discussion. According to anonymous referee 1 comments, the 4.1 section discussed the effects of swamp forest types on N2O emissions during the nongrowing season. We have rewritten the 4.1 section. According to your comments, the season specific analysis was discussed

in the 4.3 section.

2. L389, The annual N2O emissions showed significant temporal variations in grasslands. here is the focus on grasslands not very clear as you investigated swamp forests - I see that there aren't many other studies, yet you would are to explain on why this is comparable or not?

The authors' answer:

Thank you for your suggestion. To our knowledge, the N2O emissions from permafrost region were reported from peatland (Marushchak et al., 2011; Palmer & Horn, 2012; Repo et al., 2009), tundra (Lamb et al., 2011), forest (Takakai et al., 2008), grassland (Takakai et al., 2008), alpine meadow (Chen et al., 2017), thermokarst (Mu et al., 2017), beach (Zhu et al., 2012), and lake edge (Gregorich et al., 2006). Except for our previous studies, N2O emissions from swamp forests were not reported in other permafrost region (Gao et al, 2019a, 2019b). Thus, we selected the permafrost ecosystems in the permafrost region, which could be not the swamp forest.

Reference

Chen, X. P., Wang, G. X., Zhang, T., Mao, T. X., Wei, D., Hu, Z. Y., and Song, C. L.: Effects of warming and nitrogen fertilization on GHG flux in the permafrost region of an alpine meadow, Atmos Environ, 157, 111-124, https://doi.org/10.1016/j.atmosenv.2017.03.024, 2017.

Gao, W. F., Yao, Y. L., Gao, D. W., Wang, H., Song, L. Q., Sheng, H. C., Cai, T. J., and Liang, H.: Responses of N2O emissions to spring thaw period in a typical continuous permafrost region of the Daxing'an Mountains, northeast China, Atmos Environ, 214, 116822, https://doi.org/10.1016/j.atmosenv.2019.116822, 2019a.

Gao, W. F., Yao, Y. L., Liang, H., Song, L. Q., Sheng, H. C., Cai, T. J., and Gao, D. W.: Emissions of nitrous oxide from continuous permafrost region in the Daxing'an Mountains, Northeast China, Atmos Environ, 198, 34-45,

https://doi.org/10.1016/j.atmosenv.2018.10.045, 2019b.

Gregorich, E. G., Hopkins, D. W., Elberling, B., Sparrow, A. D., Novis, P., Greenfield, L. G., and Rochette, P.: Emission of $CO_2$, $CH_4$ and $N_2O$ from lakeshore soils in an Antarctic dry valley, Soil Biol Biochem, 38, 3120-3129, 10.1016/j.soilbio.2006.01.015, 2006.

Lamb, E. G., Han, S., Lanoil, B. D., Henry, G. H. R., Brummell, M. E., Banerjee, S., and Siciliano, S. D.: A High Arctic soil ecosystem resists long-term environmental manipulations, Global Change Biol, 17, 3187-3194, https://doi.org/10.1111/j.1365-2486.2011.02431.x, 2011.

Marushchak, M. E., Pitkamaki, A., Koponen, H., Biasi, C., Seppala, M., and Martikainen, P. J.: Hot spots for nitrous oxide emissions found in different types of permafrost peatlands, Global Change Biol, 17, 2601-2614, https://doi.org/10.1111/j.1365-2486.2011.02442.x, 2011.

Mu, C. C., Abbott, B. W., Zhao, Q., Su, H., Wang, S. F., Wu, Q. B., Zhang, T. J., and Wu, X. D.: Permafrost collapse shifts alpine tundra to a carbon source but reduces $N_2O$ and $CH_4$ release on the northern Qinghai-Tibetan Plateau, Geophys Res Lett, 44, 8945-8952, https://doi.org/10.1002/2017GL074338, 2017.

Palmer, K., and Horn, M. A.: Actinobacterial nitrate reducers and proteobacterial denitrifiers are abundant in $N_2O$-metabolizing palsa peat, Appl Environ Microb, 78, 5584-5596, https://doi.org/10.1128/aem.00810-12, 2012.

Repo, M. E., Susiluoto, S., Lind, S. E., Jokinen, S., Elsakov, V., Biasi, C., Virtanen, T., and Martikainen, P. J.: Large $N_2O$ emissions from cryoturbated peat soil in tundra, Nat Geosci, 2, 189-192, https://doi.org/10.1038/NGEO434, 2009.

Takakai, F., Desyatkin, A. R., Lopez, C. M. L., Fedorov, A. N., Desyatkin, R. V., and Hatano, R.: $CH_4$ and $N_2O$ emissions from a forest-alas ecosystem in the permafrost taiga forest region, eastern Siberia, Russia, J Geophys Res-Biogeosci, 113, G02002,

https://doi.org/10.1029/2007JG000521, 2008.

Zhu, R. B., Chen, Q. Q., Ding, W., and Xu, H.: Impact of seabird activity on nitrous oxide and methane fluxes from High Arctic tundra in Svalbard, Norway, J Geophys Res-Biogeosci, 117, 1703-1705, https://doi.org/10.1029/2012jg002130, 2012.

3. L392-393, The N2O was taken up during the freezing period, taken up by what - isnt that primaroly due to diffusion into snow or similar?

The authors' answer:

The mechanism of N2O absorption in marsh during the freezing period probably results from frozen soil absorbing atmospheric N2O through frost-induced cracks and perhaps the anaerobic environment was too strong and reduce N2O to N2 by reductase enzymes (Du et al., 2006).

Reference

Hao, Q. J., Wang, Y. S., Song, C. C., and Huang, Y.: Contribution of winter fluxes to the annual CH4, CO2 and N2O emissions from freshwater marshes in the Sanjiang Plain, J Environ Sci-China, 18, 270-275, https://doi.org/10.3321/j.issn:1001-0742.2006.02.012 2006.

4. L394, specified periods, here you mean seasons?

The authors' answer:

The specified periods means that the freezing period, thawing period, and growing season.

5. L395, These trends were also observed in the permafrost region. irrelevant, because you focus on permafrost regions

The authors' answer:

Thank you for your suggestion. We have deleted it.

6. L395-396, In the "hot spots" of N2O emission from permafrost region, which are those?

The authors' answer:

The "hot spots" of N2O emission from permafrost region were bare peat circles in the arctic and subarctic (Marushchak et al., 2011; Repo et al., 2009). The nongrowing season contributed by 20–69% to the annual emissions from the peat circles, which were $1.40 \pm 0.15$ g N2O m–2.

Reference

Marushchak, M. E., Pitkamaki, A., Koponen, H., Biasi, C., Seppala, M., and Martikainen, P. J.: Hot spots for nitrous oxide emissions found in different types of permafrost peatlands, Global Change Biol, 17, 2601-2614, https://doi.org/10.1111/j.1365-2486.2011.02442.x, 2011.

Repo, M. E., Susiluoto, S., Lind, S. E., Jokinen, S., Elsakov, V., Biasi, C., Virtanen, T., and Martikainen, P. J.: Large N2O emissions from cryoturbated peat soil in tundra, Nat Geosci, 2, 189-192, https://doi.org/10.1038/NGEO434, 2009.

7. L396-398, high N2O emissions were observed during the nongrowing season in the bare peatland, which contributed 20–69% to the annual emissions from the bare peatland (Marushchak et al., 2011). okay but again you are not looking at peatlands but swamp forest

The authors' answer:

Thank you for your suggestion. Except for our previous studies, N2O emissions from swamp forests were not reported in other permafrost region. Thus, we select the permafrost ecosystems from the permafrost region, which could be not the swamp forest.

8. L400, were mainly negligible, everytime and would this also be relevant for your study?

The authors' answer:

Thank you for your suggestion. We have deleted it.

9. L403-405, However, the drivers of N2O emissions between the nongrowing season and growing season were not clear in the permafrost region. says who?

The authors' answer:

This was what we concluded from previous studies. We summarized the studies on N2O emissions from permafrost regions. We found that the difference of N2O emissions between nongrowing season and growing season and their drivers was not reported before in the permafrost region.

10. L405-410, Our results showed that the N2O emissions were the highest during the growing season and lowest during the nongrowing season. The mean N2O emissions from the growing season were significantly higher than the emissions from the winter in the LL and B sites, whereas the N2O emissions during the spring thaw period was not significantly different from growing season and winter in the three swamp forests (Fig. 3). this is a repetition of the results.

The authors' answer:

Thank you for your suggestion. We deleted this part in the discussion section and integrated it to the results section. Please see lines 308-311.

11. L418-421, We found that the mean N2O emissions of the three specified periods were significantly positively correlated with soil temperature at 5, 10, and 15 cm, which could explain 91.36–94.07, 91.97–95.92, and 81.71–92.85% of the temporal variation of mean N2O emissions in the LL, LC, and B sites, respectively (Fig. 4). how or why is this different from figure 2?

The authors' answer:

In the figure 2, the linear models were used to quantify the relationship between N2O

emission and soil temperature. We found that the temporal variation of N2O emissions were significantly positively correlation with soil temperature during the nongrowing season and entire period. Thus, the soil temperature controlled the temporal variation of soil N2O emissions in the permafrost region. The Figure 4 focused on the drivers of mean N2O emissions among the three specific periods (the winter, spring thaw period, and growing season). The soil temperature drivers the difference of N2O emissions among winter, spring thaw period, and growing season. Thus, the Figure 2 and the Figure 4 was different.

12. L442, Figure 4, Which is which period? panel identifiers are mission, are these aggregated values and if so how were these aggregated? are all three soil temperature levels necessary? Which depth is the most relevant?

The authors' answer:

Thank you for your meaningful suggestion. The legend of Figure 4a should be growing season, winter, and spring thaw period. The spots with black, light gray, and gray represent the growing season, winter, and spring thaw period, respectively. We have revised it. Please see Figure 4. Yes, these were aggregated values. We calculated the average N2O emission during the growing season, winter, and spring thaw period in each base collar. And then the average N2O emission from each collar was used to compare the difference of three specific period. According to the adjust R2, the soil temperature at 5 cm depth were the most relevant with N2O emissions in the LC and B sites. While, soil temperature at 10 cm depth was the most relevant with N2O emissions in the LL sites. Thus, we think that all three soil temperature levels are necessary.

13. L448, was, were.

The authors' answer:

We have revised it.

14. L450, 14 kg ha–1, units.

The authors' answer:

Thank you for your suggestion. We replace the kg ha−1 with kg N ha−1.

15. L451-452, released approximately 0.1 Tg yr−1 N2O emissions to the atmosphere in the bare peat region of the Arctic, accounting for up to 0.6% of the global annual N2O emissions. how was the upscaling done?

The authors' answer:

Global N2O emissions from the peat circles were estimated using the mean 4% coverage of peat circles in the peat plateau, a 20% coverage of the peat plateaus in the Arctic (Walker et al., 2005; Virtanen et al., 2004) and the total land area of the tundra zone of $7:34 \times 106$ km2 (Matthews, 1983). The finally compared the global annual N2O estimate with the Arctic methane emissions (20 Tg CH4 yr−1) using the GWP approach (Christensen, 1993).

Reference

Christensen, T. R.: Methane emission from arctic tundra, Biogeochemistry, 21, 117-139, 10.1007/BF00000874, 1993.

Matthews, E.: Global vegetation and land use: new high-resolution data bases for climate studies, Journal of Applied Meteorology, 22, 474-487, https://doi.org/10.1175/1520-0450(1983)022<0474:GVALUN>2.0.CO;2, 1983.

Virtanen, T., Mikkola, K., and Nikula, A.: Satellite image based vegetation classification of a large area using limited ground reference data: a case study in the Usa Basin, north-east European Russia, Polar Res, 23, 51-66, https://doi.org/10.3402/polar.v23i1.6266, 2004.

Walker, D., Raynolds, M., Daniëls, F., Einarsson, E., Elvebakk, A., Gould, W., Katenin, A., Kholod, S., Markon, C., Melnikov, E., Moskalenko, N., Talbot, S., Yurtsev, B.,

and Team, T.: The Circumpolar Arctic vegetation map, J Veg Sci, 16, 267-282, https://doi.org/10.1111/j.1654-1103.2005.tb02365.x, 2005.

16. L455-456, However, the contribution of the nongrowing season to the annual N2O budget is uncertain in the permafrost region. is or was - after your study I thought it would be less uncertain.

The authors' answer:

Thank you for your suggestion. We have rewritten it. Please see lines 465-467.

17. L458-460, The cumulative N2O emissions during the nongrowing season ranged from 0.89 to 1.44 kg ha–1, which contributed to 41.96–53.73% of the annual budget in the permafrost region of the Daxing'an Mountains. redundant information.

The authors' answer:

Thank you for your suggestion. We have deleted the redundant information.

18. L470, frigid terrestrial ecosystems, lots of repetitions.

The authors' answer:

Thank you for your comments. We have deleted the repetition.

19. L470-472, When the pulse of N2O emissions occurred during the spring thaw period, emissions from the non-growing seasons dominated (67–74%) the annual total emissions. is this now this study or the study by Fu et al - unclear

The authors' answer:

Thank you for your suggestion. This study was by Fu et al., (2018). We have revised it.

20. L476-480, The cumulative N2O emissions during the spring thaw period ranged from 0.35 to 0.66 kg ha–1 and contributed 15.61 to 33.00% of the annual budget in the permafrost region of the Daxing'an Mountains, and these ranges were generally lower than the emissions during the winter and growing season. you state many different numbers concerning the contribution of non growing season N2O emissions to the annual budget - please clarify what is what. Also you always state the Daxing'an Mountains - however to my understanding you looked at swamp forests "only" in the region, correct?

The authors' answer:

Thank you for your suggestion. According to previous studies, the pulse of N2O emissions during the spring thaw period had significantly influence on the contribution of the nongrowing season to the total annual N2O budget (Fu et al., 2018; Li et al., 2012). We want to state that there was no pulse of N2O emissions were occurred in the present study. Thus, the contribution of nongrowing season N2O emissions to the annual budget were lower. We have rewritten it. Please see lines 504-508. Yes, it was swamp forests. We have revised it.

Reference

Fu, Y., Liu, C., Lin, F., Hu, X., Zheng, X., Zhang, W., and Ca, G.: Quantification of year-round methane and nitrous oxide fluxes in a typical alpine shrub meadow on the Qinghai-Tibetan Plateau, Agr Ecosyst Environ, 255, 27-36, 636 https://doi.org/10.1016/j.agee.2017.12.003, 2018.

Li, K. H., Gong, Y. M., Song, W., Lv, J. L., Chang, Y. H., Hu, Y. K., Tian, C. Y., Christie, P., and Liu, X. J.: No significant nitrous oxide emissions during spring thaw under grazing and nitrogen addition in an alpine grassland, Global Change Biol, 18, 2546-2554, https://doi.org/10.1111/j.1365-2486.2012.02704.x, 2012.

21. L483-484, which were not the same as the cumulative N2O emissions in the three swamp forests. why? what do you mean?

The authors' answer:

Thank you for your suggestion. We mean that the cumulative N2O emissions were not lowest in the winter and highest in the growing season as the mean N2O emissions.

Because of the length of the season had significantly affected on the cumulative N2O emissions. We have revised it. Please see lines 511-512.

22. L495-499, Our results confirmed that half of the N2O emissions were released during the nongrowing season, indicating that the N2O emissions during the nongrowing season cannot be ignored in the permafrost regions. In the future, the N2O emissions of the nongrowing season should be emphasized in the permafrost region, especially in the context of global climate change. this is the key message and the parts before are quite chaotic I must admit - thus kindly structure logically.

The authors' answer:

Thank you for your meaningful suggestion. According to your suggestion, we have revised the logical structure before this part.

23. L504, by air temperature, soil temperature, before it was stated that soil temperature was the most important driving factor

The authors' answer:

Yes, the soil temperature was the most important driving factor in our study. In this part, we summarized the driving factors of N2O emissions from the permafrost region during the growing season. We found that the N2O emissions from permafrost region were influenced by air temperature, soil temperature, water table level, soil moisture, precipitation, pH, $NH_4^+$-N, $NO_3^-$-N, TOC, gross N mineralization, N content, and C/N ratio during the growing season. We have revised it. Please see line 530-532.

24. L508-509, However, the drivers of nongrowing season N2O emission remain unknown in the permafrost region. why?

The authors' answer:

Previous studies mainly focused on the N2O emissions and their driving factors during the growing season. However, seldom studies reported the N2O emissions during

nongrowing seasons in the permafrost region. The relationship between nongrowing season N2O emissions and environment factors was not reported. Thus, the drivers of nongrowing season N2O emission remain unknown in the permafrost region.

25. L511-512, are very complex and can be produced by nitrification, nitrifier denitrification, and denitrification in the permafrost region. not just in the permafrost region. please look at Butterbach-Bahl et al 2013 PToRS for a review on the processes of N2O.

The authors' answer:

Thank you for your meaningful suggestion. We summarized the produce pathway of N2O emissions from permafrost regions. We found that the N2O emissions from permafrost region were mainly came from nitrification, denitrification, and nitrifier denitrification (Ma et al., 2007; Gil et al., 2017; Siljanen et al., 2019). Indeed, the N2O emissions could produce by nitrification, denitrification, nitrifier denitrification, coupled nitrification-denitrification, co-denitrification, and dissimilatory nitrate reduction to ammonium (Butterbach-Bahl et al., 2013). However, except for nitrification, denitrification, and nitrifier denitrification, other pathway of N2O emissions was not reported in the permafrost region. Reference

Butterbach-Bahl, K., Baggs, E. M., Dannenmann, M., Kiese, R., and Zechmeister-Boltenstern, S.: Nitrous oxide emissions from soils: how well do we understand the processes and their controls?, Philosophical Transactions of the Royal Society B Biological Sciences, 368, 20130122, 10.1098/rstb.2013.0122 2013.

Gil, J., Pérez, T., Boering, K., Martikainen, P. J., and Biasi, C.: Mechanisms responsible for high N2O emissions from subarctic permafrost peatlands studied via stable isotope techniques, Global Biogeochem Cy, 31, 172-189, https://doi.org/10.1002/2015GB005370, 2017.

Ma, W. K., Schautz, A., Fishback, L. A. E., Bedard-Haughn, A., Farrell, R. E., and Siciliano, S. D.: Assessing the potential of ammonia oxidizing bacteria to produce nitrous

oxide in soils of a high arctic lowland ecosystem on Devon Island, Canada, Soil Biol Biochem, 39, 2001-2013, https://doi.org/10.1016/j.soilbio.2007.03.001, 2007.

Siljanen, H., J.E. Alves, R., Ronkainen Jussi, G., Lamprecht, R., Bhattarai, H. R., Bagnoud, A., Marushchak, M., Martikainen, P., Schleper, C., and Biasi, C.: Archaeal nitrification is a key driver of high nitrous oxide emissions from arctic peatlands, Soil Biol Biochem, 137, 107539, https://doi.org/10.1016/j.soilbio.2019.107539, 2019.

26. L538-539, The N2O emissions were major produced by denitrification in the permafrost region. how did you determine this?

The authors' answer:

According to previous studies, the denitrification can be the predominant pathway for N2O production during the soil moisture content exceed 70% (Butterbach-Bahl et al., 2013). We have cited the reference.

Reference

Butterbach-Bahl, K., Baggs, E. M., Dannenmann, M., Kiese, R., and Zechmeister-Boltenstern, S.: Nitrous oxide emissions from soils: how well do we understand the processes and their controls?, Philosophical Transactions of the Royal Society B Biological Sciences, 368, 20130122, 10.1098/rstb.2013.0122 2013.

27. L549-552, Soil temperature was the major limiting factor related to annual N2O emissions in the permafrost region. Except for soil temperature, the N2O emissions from the permafrost region were also affected by pH, NO3−-N, TN, and C/N ratio. this section is again quite unstructured or is more like a loose list of information without the logical interconnection - thus substantial rewriting is needed

The authors' answer:

Thank you for your meaningful suggestion. We have rewritten this part. Except for soil temperature, the N2O emissions from the permafrost region were also affected by pH,

NO3−-N, TN, and C/N ratio. Voigt et al., (2020) summary the studies on nitrous oxide emissions from permafrost-affected soils, which found that the high N2O emissions in pristine permafrost regions can be expected in areas with a sparse to absent vegetation cover, a low to intermediate C:N ratio, high C and high mineral N content (in particular, NO3−-N), in combination with relatively high temperatures and a favourable moisture content. We also found that the high NO3−-N content, high TOC, and relatively high temperatures would increase the N2O emissions in the permafrost region of Daxing'an Mountains. Please see lines 579-587. Reference

Voigt, C., Marushchak, M. E., Abbott, B. W., Biasi, C., Elberling, B., Siciliano, S. D., Sonnentag, O., Stewart, K. J., Yang, Y., and Martikainen, P. J.: Nitrous oxide emissions from permafrost-affected soils, Nature Reviews Earth & Environment, 1, 420-434, https://doi.org/10.1038/s43017-020-0063-9, 2020.

Conclusions

1. L555-556, The N2O emissions from the nongrowing season were quantified in the permafrost region. its not representative for the whole permafrost region I am afraid.

The authors' answer:

Thank you for your meaningful suggestion. We have revised it. The permafrost region was limited to the Daxing'an Mountains.

2. L560-561, which cannot be ignored. In the different periods studied, N2O emissions had different key limiting factors in the permafrost region. distinguish between factor and driver variables, again its not the whole permafrost region

The authors' answer:

Thank you for your meaningful suggestion. We have replaced the factors with driver variables. We also defined the permafrost region as the permafrost region of Daxing'an Mountains.

3. L562-564, and the annual N2O emissions were driven by soil temperature, whereas the growing season N2O emissions were affected by soil temperature, water table level, and their interaction. this is not a conclusion? and whats next?

The authors' answer:

Thank you for your meaningful suggestion. Yes, this was not the conclusion. We added the conclusion after this part. Our results indicated that nongrowing season N2O emissions were an important part of the annual budget, which were of great significance for the accurate assessment of regional climate change and the impact of global climate change on permafrost region. Please see lines 600-602.

Thank you for your comments; we are happy to make additional revisions if needed.

Sincerely,

The authors,

Dr. Da-Wen Gao,

School of Environment and Energy Engineering, Beijing University of Civil Engineering and Architecture, Beijing 100044, China

E-mail address: gaodw@hit.edu.cn